# A Semantic Parsing Framework for End-to-End Time Normalization

**Xin Su**
Intel
xin.su@intel.com

**Sungduk Yu** [†]
Oracle
sungduk.yu@oracle.com

**Phillip Howard**
Thoughtworks
phillip.howard@thoughtworks.com

**Steven Bethard**
University of Arizona
bethard@arizona.edu

## Abstract

Time normalization is the task of converting natural language temporal expressions into machine-readable representations. It underpins many downstream applications in information retrieval, question answering, and clinical decision-making. Traditional systems based on the ISO-TimeML schema limit expressivity and struggle with complex constructs such as compositional, event-relative, and multi-span time expressions. In this work, we introduce a novel formulation of time normalization as a code generation task grounded in the SCATE framework, which defines temporal semantics through symbolic and compositional operators. We implement a fully executable SCATE Python library and demonstrate that large language models (LLMs) can generate executable SCATE code. Leveraging this capability, we develop an automatic data augmentation pipeline using LLMs to synthesize large-scale annotated data with code-level validation. Our experiments show that small, locally deployable models trained on this augmented data can achieve strong performance, outperforming even their LLM parents and enabling practical, accurate, and interpretable time normalization. [‡]

## 1 Introduction

Time normalization refers to the task of converting temporal expressions in natural language into machine-readable formats. For example, the phrase *"three days ago"* spoken on August 25, 2024, should be normalized to 2024-08-22. Time normalization plays a crucial role in a variety of temporal reasoning applications, including literature study [Fischer and Strötgen, 2015], question answering [Su et al., 2023], event analysis [Vossen et al., 2016], and clinical decision-making [Lin et al., 2015].

Most existing time normalization systems [Kim et al., 2020, Shwartz, 2022, Lange et al., 2023] are based on the ISO-TimeML framework [Pustejovsky et al., 2010]. These systems typically follow a two-stage pipeline: first identifying temporal expressions in text, then classifying them into predefined normalized representations (e.g., mapping "noon" to a specific time of day). While effective for standard expressions, this approach is fundamentally limited by the rigidity and restricted expressivity of the ISO-TimeML schema. As pointed out by Bethard and Parker [2016], ISO-TimeML-based systems cannot effectively handle several classes of temporal expressions:

1. **Multi-span expressions** that span across calendar units, such as *"every Monday for the past three weeks"*;

---

[†]Work done while at Intel.

[‡]Code and models are available at https://github.com/clulab/normit

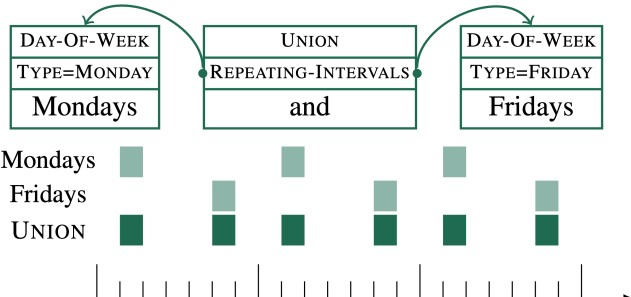

Figure 1: SCATE UNION operator example: "Mondays and Fridays" represented as the union of two repeating intervals on the timeline.

2. **Event-relative expressions**, such as *"three weeks postoperative"*, where "three weeks" is relative to "surgery";

3. **Compositional expressions** that involve multiple temporal constructs, as in *"May 22, 1995 ... and the following month"*, where the latter phrase is semantically dependent on the former.

To overcome these limitations, Bethard and Parker [2016] propose the Semantically Compositional Annotation of Temporal Expressions (SCATE) framework. SCATE represents temporal expressions through compositional and symbolic semantics using a rich set of temporal operators (e.g., Union, Intersection, RepeatingInterval). For instance, the expression *"Mondays and Fridays"* is represented as:

```
Union(RepeatingInterval(DayOfWeek(Type=Monday)),
        RepeatingInterval(DayOfWeek(Type=Friday)))
```

As illustrated in Figure 1, this operation produces all occurrences of Mondays and Fridays along the timeline. In this representation, both *Mondays* and *Fridays* are modeled as RepeatingInterval objects, and the conjunction *and* is captured using the Union operator. This compositional design allows SCATE to handle a broader range of temporal expressions and produce precisely defined, timeline-anchored intervals. Despite its expressivity, existing SCATE-based systems [Laparra et al., 2018, Xu et al., 2019] rely on complex, multi-stage pipelines. These systems treat SCATE as an annotation schema, training separate models to identify its atomic concepts and then applying handcrafted rule systems to reconstruct full interpretations. This approach results in high runtime costs, reduced maintainability, and limited deployability in real-world applications.

In this work, we propose a novel, end-to-end formulation of time normalization as a code generation task. Our key insight is to implement the full SCATE framework as an executable Python library, where all core temporal concepts and operations are mapped to Python classes and functions. This enables us to directly generate executable SCATE code from natural language inputs and deterministically compute their normalized time intervals.

Our main contributions are as follows:

- We design and implement a complete Python library that faithfully captures all concepts in SCATE, making the semantics of each time expression interpretable and executable.

- We construct detailed prompting strategies to leverage large language models (LLMs) for data augmentation, generating $10\times$ more labeled examples than existing annotated datasets, with automatic validation through code execution.

- We demonstrate that small, locally deployable models ($\leq$1B parameters) can be trained on the augmented dataset to generate SCATE code with competitive performance, enabling practical deployment of expressive time normalization systems.

## 2 Related Works

### 2.1 Time Normalization

The vast majority of existing time normalization methods are built upon the ISO-TimeML framework. Early systems, such as HeidelTime [Strötgen and Gertz, 2010] and SUTime [Chang and Manning, 2012], adopt rule-based approaches to convert recognized temporal expressions into the standardized format defined by ISO-TimeML. More recently, transformer-based models have been introduced for this task. For instance, Shwartz [2022], Lange et al. [2023], and Kim et al. [2020] train transformer-based classification models to map identified temporal expressions to predefined normalized time categories. In contrast, Laparra et al. [2018] and Xu et al. [2019] present the only complete time normalization pipeline grounded in the SCATE framework. Their approach involves using neural models, such as LSTMs, for temporal expression recognition, followed by a rule-based component to link these expressions to their final normalized forms. Distinct from all prior work, our method introduces a simple and practical end-to-end solution for time normalization formulated as a code generation task. By generating executable Python code, our approach directly maps textual temporal expressions to their normalized representations, offering a seamless and interpretable mechanism.

### 2.2 Information Extraction via Code Generation

Another related line of work gaining much recent attention is the representation of information extraction (IE) tasks using programming code or code-like structures. Instead of producing free-form text or sequences of labels, these approaches explicitly represent extracted information as structured code. This paradigm is particularly appealing in the era of powerful pretrained LLMs, which can effectively translate implicit semantic structures into explicit code-like or structured formats.

Recent studies demonstrated that code-based prompting allows efficient capturing of structured information. Code4Struct [Wang et al., 2022] and CodeIE [Li et al., 2023] framed IE tasks explicitly as code generation problems, showing that code-specialized LLMs outperform natural-language LLMs in few-shot settings for IE tasks such as named entity recognition (NER) and relation extraction (RE). Expanding this further, Code4UIE [Guo et al., 2024] introduced retrieval-augmented code prompting, retrieving relevant few-shot examples, along with their universal IE approach.

An alternative inference-time strategy by Geng et al. [2023] adapted grammar-constrained decoding, explicitly enforcing output schema via formal grammars, thus achieving competitive IE performance without any model fine-tuning.

Complementing inference-time methods, another approach focuses on training specialized IE models. Doc2Dict [Townsend et al., 2021] directly trained a generative T5 model to produce structured JSON outputs from documents, eliminating intermediate annotation steps unlike traditional pipeline-based models. More recently, KnowCoder [Li et al., 2024] proposed a dedicated IE LLM, undergoing a two-phase training (schema-based code pretraining and schema-guided instruction tuning), significantly outperforming general-purpose LLMs. In subsequent work, KnowCoder-X [Zuo et al., 2025] further extended this framework to multilingual IE tasks through cross-lingual alignment training, achieving state-of-the-art multilingual IE performance.

## 3 Methodology

**Overview** In this paper, we focus on the task of identifying temporal expressions from a given text and parsing them into the corresponding SCATE framework code. By deterministically executing the parsed structured code, we obtain time intervals anchored to a timeline. To accomplish this task, we first implement the concepts and operations defined in SCATE as fully executable Python objects packaged as a Python library, converting existing SCATE annotations into corresponding code representations. We then leverage our Python package to construct language model prompts for large-scale annotation of unlabeled text, generating additional annotations with quality-enhancing filtering mechanisms. Finally, we train small-scale language models on both the converted data and the additional LLM-annotated data for end-to-end time normalization code generation. We present an overview of our method in Figure 2.

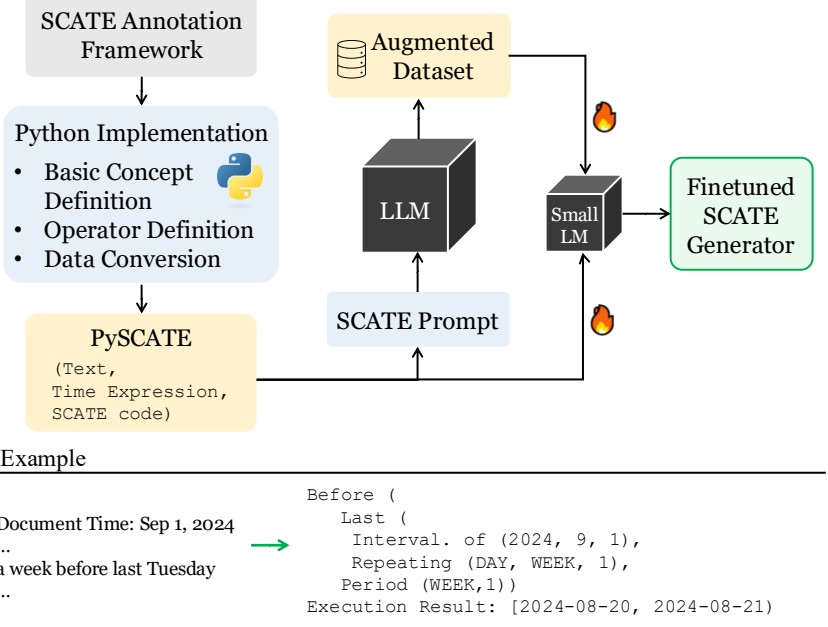

Figure 2: Overview of our approach. Top: System pipeline from Python implementation to fine-tuned model. Bottom: Example of converting "a week before last Tuesday" into executable SCATE code with normalized output.

## 3.1 Task Definition

Our proposed method is based on the SCATE temporal normalization framework. The fundamental principle of the SCATE framework is to represent complex temporal expressions compositionally, addressing the limitations of common temporal expression frameworks such as TimeML, which has limited expressivity. For instance, TimeML cannot represent expressions that cannot be aligned to a single calendar unit, such as "the past three summers."

The SCATE framework defines five key temporal concepts:

- **Timeline**: An infinite sequence of time points to which temporal expressions (or events) can be anchored. Each time point is assumed to have second-level precision, e.g., 2015-08-03 09:35:47 represents a point on the timeline.

- **Interval**: A segment that can be precisely anchored to the timeline, defined by a starting point (inclusive) and an ending point (exclusive). For example, "1990" corresponds to [1990-01-01 00:00:00, 1991-01-01 00:00:00).

- **Repeating interval**: A sequence of intervals on the timeline. For example, "Friday" refers to every Friday each week, representing a sequence of intervals that repeats infinitely.

- **Period**: An amount of time expressed as counts of standard time units. Periods are independent of the timeline. For example, "10 weeks" does not have specific start or end points.

- **Temporal operator**: A higher-order function that operates over periods, intervals, and repeating intervals to produce new temporal expressions. For example, the expression "Saturdays in March" involves two repeating intervals–"Saturdays" and "March"–combined via an INTERSECTION operator. The result is a set of Saturday intervals occurring within the month of March, which can then be anchored to the timeline. For formal operator definitions, we refer the reader to the original SCATE paper [Bethard and Parker, 2016].

In our work, we implement all SCATE temporal constructs as composable and executable Python objects. Formally, given a text $T$ containing $n$ time expressions $\{\text{timex}_1, \ldots, \text{timex}_n\}$, our goal is to

train a parameterized model $M$ that maps $T$ to a corresponding set of SCATE code representations $\{\text{code}_1, \ldots, \text{code}_n\}$ such that:

$$\text{Execute}(\text{code}_i) \rightarrow \text{Interval}_i \quad \text{for } i = 1, \ldots, n$$

where each $\text{Interval}_i$ is a normalized, timeline-anchored temporal interval.

This formulation enables the task to be approached as a structured code generation problem, with model outputs grounded in a formally defined temporal logic system.

## 3.2 SCATE Code Representation

### 3.2.1 Base Class Definitions

To faithfully represent the SCATE framework's temporal concepts, we implement a comprehensive object-oriented Python library. Our implementation centers around several base classes that directly correspond to SCATE's fundamental concepts. Our design adheres to two core principles: compositionality and executability. The compositional nature allows classes to be flexibly combined to represent complex temporal expressions, mirroring how most natural language constructs temporal references. Meanwhile, the executability principle ensures each object can be deterministically executed to produce concrete intervals on the timeline, with all classes implementing necessary addition and subtraction methods for direct interaction with Python `datetime` objects.

**Interval Class.** The Interval class serves as the cornerstone of our implementation, directly embodying SCATE's interval concept. It represents a specific time span on the timeline with well-defined start and end points. Intervals can be created through various constructors, such as `Interval.of(1990)` to represent the year 1990, `Interval.of(1990, 1, 1)` for more specific dates like January 1, 1990, or using standard ISO format through `Interval.fromisoformat`(''1990-01-01T00:00:00 1994-01-01T00:00:00'').

**Unit Class.** We implement a Python enumeration to represent standard time units (e.g., `SECOND`, `MINUTE`, `HOUR`, `DAY`, `WEEK`, `MONTH`, `YEAR`, `CENTURY`). These units serve as building blocks for periods and provide utilities for truncating dates and calculating relative deltas.

**Shift Class.** We introduce the Shift class as an abstract base class that captures the concept of movement along the timeline. The `Shift` class defines the interface for objects that can be added to or subtracted from time points to yield intervals, serving as the foundation for both SCATE's periods and repeating intervals.

**Period Class.** The Period class implements SCATE's period concept, encapsulating an amount of time expressed as counts of standard time units defined through the Unit class. For instance, "three months" is represented as `Period(MONTH, 3)`. Periods can be combined through the `PeriodSum` class to express complex durations like "two years and a day" as `PeriodSum([Period(YEAR, 2), Period(DAY, 1)])`.

**Repeating Class.** The `Repeating` class implements SCATE's repeating interval concept to capture calendar-anchored recurring time intervals. For example, "February" (all Februaries across the timeline) is represented as `Repeating(MONTH, YEAR, value=2)`, while "Thursday" would be `Repeating(DAY, WEEK, value=3)` (where we follow the `dateutil` library in using value=0 to represent Monday).We also extend the Repeating class to implement common temporal concepts as specialized classes, such as `Spring`, `Summer`, `Fall`, `Winter` for seasons, and `Morning`, `Noon`, `Afternoon`, `Evening`, `Night` for parts of the day.

### 3.2.2 Temporal Operators Class Definitions

A distinctive feature of SCATE is its rich set of temporal operators. We implement these operators through a series of Python classes.

**Positional Operator Classes.** We implement a group of positional operators primarily used to move intervals relative to reference points. Two key operator pairs are `Last`/`Next` and `Before`/`After`. The `Last` and `Next` classes respectively find the closest intervals before or after a given Shift, while `Before` and `After` move intervals backward or forward by specific time units or occurrences. For example, "the previous summer" when spoken on February 14, 1912 can be represented as `Last(Interval.of(1912, 2, 14), Summer())`, while "three Aprils after" written on January 23, 1993 can be expressed as `After(Interval.of(1993, 1, 23), Repeating(MONTH, YEAR, value=4), n=3)`. These seemingly similar but functionally distinct operators enable us to accurately capture different types of temporal expressions in natural language.

**Selection Operator Classes.** Selection operators primarily select specific instances from time sequences. The `Nth` class allows selection of the $n$-th shift from the beginning or end of an interval, while the `This` class finds the current shift containing a given interval. These operators apply to ordinal expressions (e.g., "the third Thursday") and deictic expressions (e.g., "this month"). For example, "third-to-last Sunday of 2024" can be represented as `Nth(Year(2024), Repeating(DAY, WEEK, value=6), index=3, from_end=True)`, while "this January" spoken on November 10, 1037 can be expressed as `This(Interval.of(1037, 11, 10), Repeating(MONTH, YEAR, value=1))`.

**Range Operator Classes.** Range operators handle relationships between multiple intervals. The `Between` class creates a span between two intervals, while the `Intersection` class finds the overlap among multiple intervals. These operators are particularly useful for handling time ranges and intersections of multiple temporal constraints. For example, "since 1994" written on January 9, 2007 can be represented as `Between(Year(1994), Interval.of(2007, 1, 9))`, while "earlier that day" in the context of "We met at 6:00 on January 24, 1979. Earlier that day..." would be interpreted as `Intersection([Last(Interval.of(1979, 1, 24, 6), None), Interval.of(1979, 1, 24)])`.

**Collection Operator Classes.** We implement multiple collection operators to handle sets of intervals, such as `These`, `LastN`, and `NextN`. These operators extend the functionality of basic operators, allowing us to process multiple related intervals. For example, "the next six Fridays" when written on December 22, 1714 can be represented as `NextN(Interval.of(1714, 12, 22), Repeating(DAY, WEEK, value=4), n=6)`, while "Tuesdays in January 2025" can be expressed as `These(Interval.of(2025, 1), Repeating(DAY, WEEK, value=1))`.

**Union and Intersection Classes.** We implement `ShiftUnion` and `RepeatingIntersection` classes that allow us to combine multiple shifts or find intersections of repeating intervals. For example, "Mondays and Fridays" can be represented as `ShiftUnion([Repeating(DAY, WEEK, value=0), Repeating(DAY, WEEK, value=4)])`, while "Saturdays in March" can be expressed as `RepeatingIntersection([Repeating(DAY, WEEK, value=5), Repeating(MONTH, YEAR, value=3)])`.

### 3.3 Data Augmentation

The expressive power and flexible design of SCATE introduces a potential challenge: the scarcity of large-scale annotated data. Under the original SCATE annotation framework, human annotators must identify all temporal expressions in text and their corresponding operators, as illustrated in Figure 1. This precise annotation requires domain experts; otherwise, annotation quality may be insufficient for model training. For instance, Su et al. [2021] report that even when two PhD students from related fields spent approximately 10 days on annotations, those annotations ultimately degraded model performance in temporal expression recognition due to lack of annotator training on the complex SCATE annotation guidelines.

Given our complete Python implementation of SCATE, a natural question arises: since LLMs like Claude 3.7 [Anthropic, 2025] and GPT-4.1 [OpenAI, 2025] have demonstrated unprecedented code generation capabilities, could we leverage them to identify temporal expressions in unlabeled text and generate corresponding SCATE Python code at scale? This would yield (text, time expression, python code) triplets for training smaller, more deployable language models.

To this end, we construct detailed LLM prompts (SCATE prompt) for data augmentation. We present our data augmentation prompts in markdown format, with an example shown in the Appendix B. These prompts effectively serve as formal documentation for our Python library, thoroughly introducing SCATE's key temporal concepts, defining each implemented class, detailing possible usage patterns, and providing examples to clarify potential confusion points (e.g., distinguishing between Next and After operators). Our objective is to fully leverage LLMs' code generation capabilities in a code-generation framework, adapting them to generate code for Python libraries they may not have encountered during pre-training. We iteratively develop these prompts through experimentation on the training set, refining the SCATE operator descriptions and examples to maximize LLM code generation quality.

An immediate post-generation constraint enforcement method emerges from our use of well-defined Python code objects as targets: we simply execute the generated SCATE Python code and discard samples that produce runtime errors, thus ensuring syntactically and semantically valid SCATE Python code.

## 4 Experiments

### 4.1 Datasets

TempEval-2013 [UzZaman et al., 2013] data has been annotated with publicly available SCATE annotations, including training, development, and test sets. Using our implemented SCATE Python library, we convert the original XML-formatted SCATE annotations into (sentence, time expression, SCATE code) triplets. The objective is to identify all potential time expressions in input sentences and represent them using SCATE code. To obtain a larger test set for evaluating the generalization capability of our proposed method, we merge the original development and test sets into a consolidated test set for final model performance evaluation. The resulting dataset includes 557 annotated SCATE code block in the training set and 313 in the test set. We use the training set to refine prompts and select the optimal LLM for data augmentation.

**Evaluation Metrics**   Our evaluation is based on gold-standard triplets, where each triplet consists of (sentence, time expression span, SCATE code). For each triplet, we assess the time expression by first determining whether it was successfully identified. If a time expression remains undetected, its normalization accuracy is assigned a value of 0. For identified expressions, we execute the predicted SCATE code to generate normalized time intervals and compare them with the gold-standard intervals. We assign an accuracy of 1 for exact matches and 0 otherwise. Based on this approach, we calculate standard evaluation metrics including average execution accuracy, precision, recall, and F1 score.

### 4.2 Implementation Details

**Models and Hyperparameters**   We access various LLMs through cloud-based APIs: GPT-4.1 via Azure OpenAI, Claude 3.7 via Amazon Bedrock, and Gemini [Google DeepMind, 2025b,a] via Google Cloud Platform. For local model training, we train both Qwen/Qwen2.5-0.5B-Instruct [Team, 2024] and T5-Large [Raffel et al., 2020] on a single NVIDIA 80GB A100 GPU with 5 epochs and batch size 64. The learning rates are $2 \times 10^{-5}$ for Qwen and $5 \times 10^{-5}$ for T5. We prompt or train models to generate JSON strings where each item represents a time expression and SCATE code pair, for example: `["{time_text": "recent years", "scate": "Last(interval=Interval.of(1998, 2, 13), shift=Period(unit=YEAR, n=None))}"]`.

**Data Augmentation Text**   We randomly sample 10k sentences from the CC-News [Mackenzie et al., 2020] dataset—widely used in large language model pretraining—as our source for data augmentation. It comes from the same newswire domain as with TempEval SCATE-annotated data. We then apply Claude 3.7 to these sentences using our designed SCATE prompt to generate temporal annotations. After prompt-based generation, we apply a runtime filtering step to discard syntactically invalid or semantically incoherent outputs. This process yields a total of 8,583 valid SCATE code blocks.

Table 1: Performance of LLMs on Temporal Expression Recognition and SCATE Code Generation on Training Set (557 examples).

| Model | Accuracy | Precision | Recall | F1 |
|---|---|---|---|---|
| Claude 3.5 | 0.62 | **0.64** | 0.62 | 0.63 |
| Claude 3.7 | **0.69** | 0.63 | 0.69 | **0.66** |
| Gemini 2.0 Flash | 0.64 | **0.64** | 0.64 | 0.64 |
| Gemini 2.5 Flash | 0.61 | 0.63 | 0.61 | 0.62 |
| Gemini 2.5 Pro | 0.50 | **0.64** | 0.50 | 0.56 |
| GPT-4.1 | 0.67 | 0.60 | **0.67** | 0.63 |
| Average | 0.62 | 0.63 | 0.62 | 0.62 |

Table 2: Performance comparison of different methods on the test set (313 examples).

| Methods | Accuracy | Precision | Recall | F1 |
|---|---|---|---|---|
| Qwen2.5-0.5B + Training Set | 0.01 | 1.00 | 0.01 | 0.01 |
| Qwen2.5-0.5B + CC-News | 0.37 | 0.46 | 0.37 | 0.41 |
| Qwen2.5-0.5B + CC-News + Training Set | **0.59** | **0.59** | **0.59** | **0.59** |
| T5-Large + CC-News + Training Set | 0.37 | 0.52 | 0.37 | 0.43 |
| Claude 3.7 + SCATE Prompt | 0.49 | 0.56 | 0.49 | 0.52 |
| Claude 3.7 + Interval Few-shot Prompt | 0.38 | 0.38 | 0.38 | 0.38 |
| GPT 4.1 + SCATE Prompt | 0.51 | 0.51 | 0.51 | 0.51 |

## 4.3 Main Results

**Can LLMs identify time expressions and parse corresponding SCATE code?** To answer this question and identify the optimal state-of-the-art model for our designed data augmentation, we test the most popular large language models on the training set, including Claude 3.5 Haiku, Claude 3.7 Sonnet, Gemini 2.0 Flash, Gemini 2.5 Flash, Gemini 2.5 Pro, and GPT 4.1. We prompt these models using the approach described in Section 3.3 to identify temporal expressions and generate corresponding SCATE code. The results are presented in Table 1. These state-of-the-art models achieve an average accuracy and F1 score of 0.62, indicating that while large models can generate SCATE code through prompting to a reasonable degree, there remains significant room for improvement. Among the evaluated models, Claude 3.7 demonstrates the best performance.

**Is targeting SCATE code better than directly generating time intervals?** One of our key hypotheses for time normalization is that existing LLMs can more effectively generate code representations of temporal operations than implicitly perform these operations to directly identify time expressions and their corresponding intervals. Similar to recent work [Wei et al., 2022] finding that generating chain-of-thought reasoning before providing answers significantly improves performance in mathematical reasoning, we posit that SCATE code acts as a form of chain-of-thought, with the additional benefit that we need not rely on the model to produce the final answer since we can determine this through code execution.

To validate this hypothesis, we employ conventional few-shot prompting (Interval Few-shot Prompt) to direct Claude 3.7 in identifying temporal expressions and their corresponding time intervals in input text (the specific Interval Few-shot Prompt is provided in Appendix C). We contrast this with prompting Claude 3.7 to generate SCATE code using our approach described in Section 3.3. As shown in Table 2, the SCATE code generation approach significantly outperforms direct time interval generation, surpassing it by more than 10 points in both average accuracy and F1 scores.

**Can we train a smaller local model for SCATE generation?** Temporal normalization typically functions as one component within larger data processing or retrieval pipelines. Consequently, there is a clear need for efficient solutions with low computational costs. Relying entirely on large models like Claude 3.7 presents challenges for scaling temporal normalization deployments. Therefore, a

smaller model ($\leq$ 1B parameters) capable of inference on a single consumer-grade GPU better aligns with practical requirements.

We explore this possibility by fine-tuning Qwen2.5-0.5B on three different data combinations: on the original TempEval 2013 training set (Training Set), on data augmentation-labeled data (CC-News), and on a combination of both (CC-News + Training Set). We observe that small models struggle to achieve reasonable performance on limited datasets, even with manually annotated training data from TempEval, with average accuracy and F1 scores approaching zero. Fine-tuning on augmentation-labeled data shows notable improvement, though still maintaining a considerable gap compared to its parent data augmentation method (Claude 3.7 + SCATE Prompt).

However, when we combine our augmented data with the original training set, we observe significant performance improvements, achieving 0.59 average accuracy on the test set—surpassing its parent data augmentation method (Claude 3.7 + SCATE Prompt) by 10 points. This demonstrates the complementary nature of augmented and original training data, and confirms that smaller, deployable models can effectively perform temporal normalization when provided with sufficient and diverse training examples.

**How efficient is our method compared to existing systems?**    Beyond accuracy improvements, our method offers significant computational advantages. On the test set, our fine-tuned Qwen2.5-0.5B model achieves 31.7× faster inference than the previous LSTM-based system [Xu et al., 2019], processing the entire test set in 6 seconds compared to 190 seconds. Additional comparisons and statistical significance analysis with 95% confidence intervals are provided in Appendix D.

**How do encoder-decoder models compare to decoder-only architectures?**    To validate our choice of decoder-only architecture, we also fine-tune T5-Large, a representative encoder-decoder model widely used for sequence-to-sequence tasks. As shown in Table 2, T5-Large achieves an F1 score of 0.43, comparable to our Qwen2.5-0.5B decoder-only model trained only on CC-News (F1: 0.41) but significantly lower than the full Qwen model (F1: 0.59). This demonstrates that decoder-only architectures are better suited for our SCATE code generation formulation, with Qwen2.5-0.5B achieving superior performance despite having fewer parameters (500M vs. 770M).

**What are the main errors in the best-performing fine-tuned model?**    We conducted an error analysis on 20 errors from the Qwen2.5-0.5B + CC-News + Training Set model. The analysis reveals that the most common issue is missed temporal expressions (70%), where the system fails to identify time phrases annotated in the gold annotations. The second most frequent problem is boundary errors (10%), where the system's identified temporal expression has boundary differences from the gold standard, despite the SCATE expression being logically similar. For example, the gold standard might annotate "year," while the system identifies "within a year."

Some errors (10%) are structural, such as in "third-quarter net loss... year-earlier," which involves nested temporal comparisons (using Before(Nth(...)) structure), but the system only captures individual fragments rather than the complete structure. SCATE type errors (5%) manifest as inappropriate operator selection, such as labeling "later" as Next(...) instead of the more semantically appropriate After(...). Granularity errors (5%) occur when the system represents specific expressions with coarser time units, such as simplifying "11/02/89" to Year(1989) instead of preserving the month and day information.

Overall, the primary challenges lie not in the model's ability to generate SCATE code but in span recognition—a traditional NLP task. Improving the model's identification of temporal expression spans represents a promising research direction for enhancing overall performance, or alternatively, introducing an additional small classification model specifically for span identification.

# 5   Limitations

We focus on TempEval-2013, the only publicly available dataset with full SCATE annotations, as the primary benchmark for evaluation. While this dataset provides high-quality supervision, it does not fully capture the diversity of temporal expressions found in open-domain or multilingual scenarios. Our model fine-tuning experiments are limited to a single small-scale open-source model (Qwen2.5-0.5B), without exploring alternative architectures or larger models. Additionally, our data

augmentation pipeline relies on a subset of proprietary LLMs, and we do not systematically compare across the full range of commercial or open-source models.

# 6 Conclusion

We present a new end-to-end approach for time normalization by framing it as a code generation problem based on the SCATE framework. Our method unifies symbolic temporal semantics with executable representations, enabling deterministic and interpretable normalization of complex time expressions. Through comprehensive Python implementation and carefully designed prompting strategies, we show that LLMs can effectively identify temporal expressions and generate high-quality SCATE code. More importantly, we demonstrate that fine-tuning small models on a combination of LLM-augmented and human-annotated data achieves strong performance while remaining deployable on standard hardware. Our findings suggest that compositional code generation offers a scalable and semantically grounded solution for time normalization.

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

## A AI Use Declaration

Code was developed with support from GitHub Copilot. ChatGPT was used for editing for grammar and clarity in some sections.

## B SCATE Prompt

# SCATE Time Normalization in Python

## Introduction

Time normalization is the task of translating natural language expressions of time into a standardized representation. The SCATE (Semantically Compositional Annotation of Time Expressions) framework represents different time expressions in a compositional manner, formally defining the semantics of various time expressions and representing the normalized time expressions as mathematical operations on intervals on a timeline.

## Key Time Concepts

| Concept | Description |
|---------|-------------|
| Timeline | The infinite sequence of time points to which time expressions should be anchored. Each time point marks the start of a second (e.g., 2015-08-03 09:35:47). |
| Interval | An interval on the timeline, defined by a starting point (inclusive) and an ending point (exclusive). E.g., "1990" corresponds to [1990-01-01, 1991-01-01). |
| Repeating interval | A sequence of intervals on the timeline. E.g., "Friday" refers to every Friday each week, a sequence of intervals that repeats infinitely. |
| Period | An amount of time, expressed as counts of standard time units. Periods are independent of the timeline. E.g., "10 weeks" doesn't have specific start/end points. |
| Temporal operator | A function that takes periods, intervals, and/or repeating intervals as arguments and produces new periods, intervals, or repeating intervals. |

## Basic Usage

When using the SCATE library, the basic approach is to:

1. Identify time expressions in text

2. Generate the corresponding SCATE Python code for those expressions

## API Reference

### Interval Classes

```
class Interval(start:datetime, end:datetime) -> Interval
```

Implements the SCATE interval definition. Defined by a starting point (inclusive) and an ending point (exclusive).

**Example:**

```
# Representing the year 1990
Interval(
    start=datetime.datetime.fromisoformat("1990-01-01T00:00:00"),
    end=datetime.datetime.fromisoformat("1991-01-01T00:00:00")
)
```

## Methods

```
@classmethod
Interval.fromisoformat(string:str) -> Interval
```

Creates an Interval from two dates in ISO 8601 format.

**Example:**

```
# Representing May 1362
Interval.fromisoformat("1362-05-01T00:00:00 1362-06-01T00:00:00")
# Or more concisely
Interval.fromisoformat("1362-05-01 1362-06-01")
```

```
@classmethod
Interval.of(*args:int) -> Interval
```

Creates an Interval that aligns to exactly one calendar unit.

**Parameters:**

- `*args (int)` – A starting point specified by any prefix of the list: year, month, day, hour, minute, second, microsecond.

**Examples:**

```
# Representing the year 1990
Interval.of(1990)

# Representing April 1, 1918
Interval.of(1918, 4, 1)
```

## Intervals Class

```
class Intervals(abc.Iterable[Interval])
```

A collection of intervals on the timeline. This class implements the Python Iterable interface to allow iteration over a set of Interval objects.

**Usage:**
This class is typically returned by operations that produce multiple intervals, such as `LastN`, `NextN`, `NthN`, and `These`.

## Unit Classes

```
class Unit(enum.Enum)
```

A named unit of time.

**Available units:**

- MICROSECOND
- MILLISECOND
- SECOND
- MINUTE
- HOUR
- DAY
- WEEK
- MONTH
- QUARTER_YEAR
- YEAR
- DECADE
- QUARTER_CENTURY
- CENTURY

### Methods

```
Unit.truncate(dt:datetime.datetime) -> datetime.datetime
```

Sets all units smaller than this one in the datetime to zero.

```
Unit.relativedelta(n) -> dateutil.relativedelta.relativedelta
```

Constructs a relativedelta object representing a number of repetitions of this unit.

```
Unit.expand(interval:Interval, n:int=1) -> Interval
```

Expands an interval to the width of a number of repetitions of this unit.

## Shift Classes

```
class Shift
```

Base class for objects that can be added or subtracted from a time point yielding an Interval.

```
class Period(unit:Unit, n:int|None) -> Shift
```

Implements the SCATE Period definition. An amount of time, expressed as counts of standard time units.

**Example:**

```
# Representing "three months"
Period(MONTH, 3)
```

```
class PeriodSum(periods:list[Period]) -> Shift
```

A period whose duration is the sum of two or more periods.

**Example:**

```
# Representing "two years and a day"
PeriodSum([Period(YEAR, 2), Period(DAY, 1)])
```

## Repeating Interval Classes

```
class Repeating(unit:Unit, range:Unit=None, value:int=None, n_units:int=1) -> Shift
```

Implements the SCATE repeating interval definition. A Repeating identifies intervals that are named by the calendar system and repeat along the timeline.

**Examples:**

```
# Representing all months of "February"
Repeating(MONTH, YEAR, value=2)

# Representing all generic calendar "day"
Repeating(DAY)

# Representing "Thursday" (Monday=0, so Thursday=3)
Repeating(DAY, WEEK, value=3)
```

**Predefined Repeating Intervals**

| Class | Description |
|-------|-------------|
| Spring | Meteorological springs in Northern Hemisphere (March-May) |
| Summer | Meteorological summers in Northern Hemisphere (June-August) |
| Fall | Meteorological falls in Northern Hemisphere (September-November) |
| Winter | Meteorological winters in Northern Hemisphere (December-February) |
| Weekend | Weekends (Saturdays and Sundays) |
| Morning | Meteorological mornings, 06:00 until 12:00 |
| Noon | Noons, 12:00 until 12:01 |
| Afternoon | Meteorological afternoons, 12:00 until 18:00 |
| Day | Meteorological daytime, 06:00 until 06:00 |
| Evening | Meteorological evenings, 18:00 until 00:00 |
| Night | Meteorological nights, 00:00 until 06:00 |
| Midnight | Midnights, 00:00 until 00:01 |

## Composite Interval Classes

### EveryNth Class

```
class EveryNth(shift:Shift, n:int) -> Shift
```

Creates a repeating interval that selects only every nth occurrence of another shift pattern.

**Parameters:**

- `shift` - The base shift pattern
- `n` - How many occurrences to skip between each selected occurrence

**Example:**

```
# Representing "every other Friday"
EveryNth(Repeating(DAY, WEEK, value=4), n=2)
```

### ShiftUnion Class

```
class ShiftUnion(shifts:Iterable[Shift]) -> Shift
```

Creates a union of two or more shift patterns, including all occurrences from each pattern.

**Parameters:**

- `shifts` - A collection of shift patterns to combine

**Example:**

```
# Representing "Mondays and Fridays"
ShiftUnion([
    Repeating(DAY, WEEK, value=0),  # Monday
    Repeating(DAY, WEEK, value=4)   # Friday
])
```

### RepeatingIntersection Class

```
class RepeatingIntersection(shifts:Iterable[Repeating]) -> Shift
```

Creates an intersection of two or more repeating intervals, including only times that satisfy all patterns.

**Parameters:**

- `shifts` - A collection of repeating intervals to intersect

**Example:**

```
# Representing "Saturdays in March"
RepeatingIntersection([
    Repeating(DAY, WEEK, value=5),   # Saturday
    Repeating(MONTH, YEAR, value=3)  # March
])
```

## Year-specific Classes

```
class Year(digits:int, n_missing_digits:int=0) -> Interval
```

The interval from the first second of a year to the first second of the next year. With `n_missing_digits` parameter, it can also represent longer intervals such as decades, centuries, or millennia.

**Parameters:**

- `digits` - The significant digits of the year or time period
- `n_missing_digits` - The number of trailing digits omitted, determining the granularity:
  - 0: represents a single year (e.g., 2014), from the first second of the year to the first second of the next year
  - 1: represents a decade (e.g., 201X or the 2010s), spanning 10 years
  - 2: represents a century (e.g., 20XX or the 2000s), spanning 100 years
  - 3: represents a millennium (e.g., 2XXX or the 2000s millennium), spanning 1000 years

**Examples:**

```
# Representing the year 2014
Year(2014)  # [2014-01-01T00:00:00, 2015-01-01T00:00:00)

# Representing the decade of the 1980s
Year(198, n_missing_digits=1)  # [1980-01-01T00:00:00, 1990-01-01T00:00:00)

# Representing the 20th century
Year(19, n_missing_digits=2)   # [1900-01-01T00:00:00, 2000-01-01T00:00:00)

# Representing the 2nd millennium CE
Year(1, n_missing_digits=3)    # [1000-01-01T00:00:00, 2000-01-01T00:00:00)
```

```
class YearSuffix(interval:Interval, digits:int, n_missing_digits:int=0) -> Interval
```

A year-long interval (or longer with `n_missing_digits`) created from the year of another interval and a suffix of digits to replace in that year. Like the `Year` class, with `n_missing_digits` it can represent decades, centuries, etc.

**Parameters:**

- `interval` - The reference interval containing the base year
- `digits` - The digits to use as the suffix of the year
- `n_missing_digits` - The number of trailing digits omitted, determining the granularity:
    - 0: represents a single year
    - 1: represents a decade
    - 2: represents a century
    - etc.

**Examples:**

```
# Representing the year "96" in the context of 1993 (becomes 1996)
YearSuffix(Year(1993), digits=96)  # [1996-01-01T00:00:00, 1997-01-01T00:00:00)

# Representing the decade "20s" in the context of 1993 (becomes 1920s)
YearSuffix(Year(1993), digits=2, n_missing_digits=1)  # [1920-01-01T00:00:00, 1930-01-
01T00:00:00)
```

**Ordinal Centuries & Anniversaries**

| Text | SCATE |
|---|---|
| 20th century | `Nth(interval=None, shift=Repeating(unit=CENTURY, range=CENTURY), index=20)` |
| 100th anniversary | `Nth(interval=None, shift=Period(unit=YEAR, n=None), index=100)` |

Use `Nth` for ordinal references to long-span intervals. If no explicit anchor interval is present, set `interval=None` and resolve contextually in a post-processing step.

## Temporal Operators

### Last Operators

```
class Last(interval:Interval, shift:Shift, interval_included:bool=False) -> Interval
```

The closest preceding interval matching the specified Shift.

**Example with Period:**

```
# Representing "over the past four days" (in the context of 1 Nov 2024)
Last(Interval.of(2024, 11, 1), Period(DAY, 4))
```

**Example with Repeating:**

```
# Representing "Tuesday" (in the context of Tue 8 Nov 2016)
# The speaker references the current day
Last(
    Interval.of(2016, 11, 8),
    Repeating(DAY, WEEK, value=1),
    interval_included=True
)
```

```
class LastN(interval:Interval, shift:Shift, n:int, interval_included:bool=False) ->
Intervals
```

Repeats the `Last` operation n times.

**Example:**

```
# Representing "the previous two summers" (in the context of 29 May 1264)
LastN(Interval.of(1264, 5, 29), Summer(), n=2)
```

### Next Operators

```
class Next(interval:Interval, shift:Shift, interval_included:bool=False) -> Interval
```

The closest following interval matching the specified Shift.

**Example with Period:**

```
# Representing "the next three hours" (in the context of 1 Nov 2024)
Next(Interval.of(2024, 11, 1), Period(HOUR, 3))
```

**Example with Repeating:**

```
# Representing "the coming week" (in the context of 14 Feb 1912)
Next(Interval.of(1912, 2, 14), Repeating(WEEK))
```

```
class NextN(interval:Interval, shift:Shift, n:int, interval_included:bool=False) ->
Intervals
```

Repeats the `Next` operation n times.

**Example:**

```
# Representing "the next six Fridays" (in the context of Sat 22 Dec 1714)
NextN(Interval.of(1714, 12, 22), Repeating(DAY, WEEK, value=4), n=6)
```

## After Operators

```
class After(interval:Interval, shift:Shift, n:int=1, interval_included:bool=False) ->
Interval
```

Moves the input Interval later by the specified Shift the specified number of times.

**Example with Period:**

```
# Representing "a month later" (in the context of 13 Sep 1595)
After(Interval.of(1595, 9, 13), Period(MONTH, 1))
```

**Example with Repeating:**

```
# Representing "three Aprils after" (in the context of Sat 23 Jan 1993)
After(
    Interval.of(1993, 1, 23),
    Repeating(MONTH, YEAR, value=4),
    n=3
)
```

## Important Differences Between Next and After

There are important behavioral differences between the `Next` and `After` operators:

**With Periods:**
They behave very differently: `After` moves the reference interval by the specified amount, while `Next`
takes the period following the interval:

```
>>> After(Interval.of(2024, 4, 12), Period(WEEK, 3)).isoformat()
'2024-05-03T00:00:00 2024-05-04T00:00:00'
>>> Next(Interval.of(2024, 4, 12), Period(WEEK, 3)).isoformat()
'2024-04-13T00:00:00 2024-05-04T00:00:00'
```

**With Repeating Intervals and n=1 (default):**

They produce identical results:

```
>>> After(Interval.of(2024, 4, 12), Repeating(WEEK)).isoformat()
'2024-04-16T00:00:00 2024-04-23T00:00:00'
>>> Next(Interval.of(2024, 4, 12), Repeating(WEEK)).isoformat()
'2024-04-16T00:00:00 2024-04-23T00:00:00'
```

**With Repeating Intervals and n>1:**

They diverge significantly: `After` returns a single interval (the nth one), while `NextN` returns n intervals:

```
>>> After(Interval.of(2024, 4, 12), Repeating(WEEK), n=3).isoformat()
'2024-04-30T00:00:00 2024-05-07T00:00:00'
>>> NextN(Interval.of(2024, 4, 12), Repeating(WEEK), n=3).isoformats()
['2024-04-16T00:00:00 2024-04-23T00:00:00', '2024-04-23T00:00:00 2024-04-30T00:00:00',
 '2024-04-30T00:00:00 2024-05-07T00:00:00']
```

Note: In the pyscate/normit.time implementation, you need to use `NextN` instead of `Next` to access the `n` argument functionality.

## Before Operators

```
class Before(interval:Interval, shift:Shift, n:int=1, interval_included:bool=False) ->
Interval
```

Moves the input Interval earlier by the specified Shift the specified number of times.

**Example with Period:**

```
# Representing "a year ago" (in the context of 13 Sep 1595)
Before(Interval.of(1595, 9, 13), Period(YEAR, 1))
```

**Example with Repeating:**

```
# Representing "two Tuesdays before" (in the context of Sat 23 Jan 1993)
Before(
    Interval.of(1993, 1, 23),
    Repeating(DAY, WEEK, value=1),
    n=2
)
```

## Nth Operators

```
class Nth(interval:Interval, shift:Shift, index:int, from_end:bool=False) -> Interval
```

Selects the nth repetition of a Shift starting from one end of the Interval.

**Examples:**

```
# Representing "fiftieth day of 2016"
Nth(Year(2016), Repeating(DAY), index=50)

# Representing "Third-to-last Sunday of 2024"
Nth(Year(2024), Repeating(DAY, WEEK, value=6), index=3, from_end=True)
```

```
class NthN(interval:Interval, shift:Shift, index:int, n:int, from_end:bool=False) ->
Intervals
```

Selects a specified number of nth repetitions of a Shift starting from one end of the Interval.

**Example:**

```
# Representing "the second six Mondays of 1997"
NthN(Year(1997), Repeating(DAY, WEEK, value=0), index=2, n=6)
```

## This Operators

```
class This(interval:Interval, shift:Shift) -> Interval
```

Finds the interval containing the given interval based on the specified Shift.

**Example with Period:**

```
# Representing "these six days" (in the context of 29 Apr 1176)
This(Interval.of(1176, 4, 29), Period(DAY, 6))
```

**Example with Repeating:**

```
# Representing "this January" (in the context of 10 Nov 1037)
This(Interval.of(1037, 11, 10), Repeating(MONTH, YEAR, value=1))
```

## These Operator

```
class These(interval:Interval, shift:Shift) -> Intervals
```

Finds all instances of the specified shift pattern within the given interval. This operator first identifies the containing range of the specified shift type, then finds all individual shift units within that range.

**Parameters:**

- `interval` - The containing interval to search within
- `shift` - The shift pattern to find instances of

**Example:**

```
# Representing "Tuesdays and Thursdays in January 2025"
These(
    Interval.of(2025, 1),
    ShiftUnion([
        Repeating(DAY, WEEK, value=1),  # Tuesday
        Repeating(DAY, WEEK, value=3)   # Thursday
    ])
)
```

## Between Operator

```
class Between(start_interval:Interval, end_interval:Interval, start_included:bool=False,
end_included:bool=False) -> Interval
```

Selects the interval between a start and an end interval. This creates a new interval that spans from the start interval to the end interval.

**Parameters:**

- `start_interval` - The beginning interval
- `end_interval` - The ending interval
- `start_included` - Whether to include the start interval in the result
- `end_included` - Whether to include the end interval in the result

**Example:**

```
# Representing "since 1994" (in the context of 09 Jan 2007)
Between(Year(1994), Interval.of(2007, 1, 9))
```

### Quick Templates

| Pattern | SCATE Template |
|---------|----------------|
| since X | `Between(start_interval=X, end_interval=DCT)` |
| weeks before | `Last(interval=DCT, shift=Period(unit=WEEK, n=None))` |

Use these templates for constructions such as *since 1985*, *since March*, or *weeks before the attack*.

## Intersection Operator

```
class Intersection(intervals:Iterable[Interval]) -> Interval
```

Selects the interval in which all given intervals overlap. This creates a new interval that represents the common timespan where all input intervals coincide.

**Parameters:**

- `intervals` - A collection of intervals to find the intersection of

**Example:**

```
# Representing "earlier that day"
# (in the context of "We met at 6:00 on 24 Jan 1979")
Intersection([
    Last(Interval.of(1979, 1, 24, 6), None),
    Interval.of(1979, 1, 24)
])
```

# Time Expression Annotation Examples

Once you understand the SCATE framework, you can start annotating time expressions in text. The annotation process involves:

1. Receiving the document creation time (DCT) and text as input

2. Identifying time expressions in the text

3. Writing the corresponding SCATE code for each expression

## Financial and Reporting Period Time Expressions

Financial reports, earnings statements, and business news often contain specialized time expressions related to fiscal periods:

1. **Quarter references in financial contexts**:

| Expression | Context | Correct Representation | Incorrect Representation |
|---|---|---|---|
| "third-quarter" | Past context | `Before(interval=Nth(interval=Year(digits=YEAR), shift=Repeating(unit=QUARTER_YEAR, range=QUARTER_YEAR), index=3), shift=Period(unit=YEAR, n=None))` | `This(interval=Year(digits=YEAR), shift=Repeating(unit=QUARTER_YEAR, range=YEAR, value=3))` |
| "Q2 results" | Current | `This(interval=Year(digits=YEAR), shift=Repeating(unit=QUARTER_YEAR, range=YEAR, value=2))` | - |
| "next quarter" | Future | `Next(interval=DCT, shift=Repeating(unit=QUARTER_YEAR, range=QUARTER_YEAR))` | - |

2. **Interpreting contextual time cues**:
   - Pay attention to verb tense ("reported", "said", "announced") which may indicate the quarter reference is to a past period
   - Look for comparative language ("year-ago quarter", "previous quarter") which specifies relative time positioning
   - Check for explicit fiscal year indicators ("FY2023 Q3", "third quarter of fiscal 2022")

3. **Year-ago comparisons**:
   - Expressions like "year-ago quarter" or "same quarter last year" should use `Before` with a period of a year
   - Example: `Before(interval=This(interval=DCT, shift=Repeating(unit=QUARTER_YEAR, range=QUARTER_YEAR)), shift=Period(unit=YEAR, n=1))`

4. **Key principles for financial time expressions**:
   - Financial quarters have specific calendar alignments (Q1: Jan-Mar, Q2: Apr-Jun, etc.)

- The fiscal year may differ from the calendar year depending on the company
- Always consider whether the reference is to the current reporting period or a comparison to a previous period
- Use the appropriate operator (`This`, `Before`, `Next`) based on the temporal context of the statement

## Temporal Adverbs and Their Appropriate Operators

When annotating temporal adverbs, it's crucial to select the correct operator to accurately represent the time relation:

| Temporal Expression Category | Examples | Appropriate Operator | Example Code |
|---|---|---|---|
| Past References | previously, formerly, earlier, before, in the past, ago | `Last` | `Last(interval=DCT, shift=None)` |
| Present References | now, currently, presently, at present | `This` | `This(interval=DCT, shift=None)` |
| Future References | soon, later, in the future | `Next` | `Next(interval=DCT, shift=None)` |

**Examples of correct annotation for temporal adverbs:**

```
# For "previously" in a text with DCT of 2021-05-15
Last(interval=Interval.of(2021, 5, 15), shift=None)

# For "currently" in a text with DCT of 2021-05-15
This(interval=Interval.of(2021, 5, 15), shift=None)

# For "later" in a text with DCT of 2021-05-15
Next(interval=Interval.of(2021, 5, 15), shift=None)
```

**Comparative Adverb + Named Day**

| Text Span | Correct SCATE |
|---|---|
| earlier Sunday | `Intersection([Before(interval=DCT, shift=None), This(interval=DCT, shift=Repeating(unit=DAY, range=WEEK, value=6))])` |
| later Monday | `Intersection([After(interval=DCT, shift=None), This(interval=DCT, shift=Repeating(unit=DAY, range=WEEK, value=0))])` |
| prior year | `Before(interval=DCT, shift=Repeating(unit=YEAR, range=YEAR))` |

When a comparative adverb (*earlier, later, prior, following*, etc.) is immediately followed by a **named calendar unit** (day, month, year), treat the whole phrase as **one time expression** and combine the semantics with `Intersection` (or `Before` / `After`). Never annotate the adverb and the named unit separately.

## Contextual Time Reasoning and Future Date Inference

When annotating time expressions, particularly in business, financial, and legal contexts, it's crucial to correctly infer whether a date refers to the current year or a future year:

1. **Future event indicators**:
   - Words and phrases that signal future events: "payable", "due", "scheduled", "will be", "upcoming"
   - Look for modal verbs indicating future actions: "will", "shall", "is to be"

2. **Date logic with Document Creation Time (DCT)**:

| Expression | Context | DCT | Correct Interpretation | Correct Representation |
|---|---|---|---|---|
| "Jan. 2" | "payable Jan. 2" | November 1989 | January 2, 1990 | `Next(interval=DCT, shift=RepeatingIntersection([Repeating(unit=MONTH, range=YEAR, value=1), Repeating(unit=DAY, range=MONTH, value=2)]))` |
| "Dec. 15" | "record Dec. 15" | November 1989 | December 15, 1989 | `Next(interval=DCT, shift=RepeatingIntersection([Repeating(unit=MONTH, range=YEAR, value=12), Repeating(unit=DAY, range=MONTH, value=15)]))` |

   - "last <Month>" → If **Month ≥ DCT.month**, the reference is to that month **in the previous year**; otherwise it is the most recent occurrence within the current year.
   - "next <Month>" → If **Month ≤ DCT.month**, the reference is to that month **in the next year**; otherwise it is the upcoming occurrence within the current year.
   - Example (DCT = 1998-03-01): "last February" → 1997-02 (previous year) , "next February" → 1999-02 (next year)

3. **Month sequence reasoning**:
   - If a month mentioned is earlier in the calendar than the DCT month, and context suggests a future event, it typically refers to that month in the following year
   - Example: If DCT is November 1989, then "Jan. 2" likely refers to January 2, 1990, not January 2, 1989

4. **Common patterns in financial announcements**:
   - Dividend/payment announcements typically include:
     - A declaration date (usually the DCT or before)
     - A record date (usually in the near future)
     - A payment date (usually after the record date)
   - Example: "The dividend is payable [payment date] to shareholders of record on [record date]"

5. **Incorrect annotation patterns to avoid**:
   - Never default to `This(interval=Year(digits=YEAR), ...)` for dates without explicitly considering whether they refer to the current or future year

- Don't assume months mentioned in text always refer to the current year
- Pay attention to the logical sequence of events implied by the text

## Handling Prepositions in Time Expressions

When annotating time expressions, pay careful attention to whether prepositions (like "over", "during", "within", etc.) should be included as part of the time expression or treated separately:

1. **Separate core temporal expressions from prepositions when appropriate**:
   - For expressions like "over the weekend", the core temporal expression is often just "weekend"
   - The preposition "over" typically provides context but isn't part of the time expression itself

2. **Examples of proper boundary detection with prepositions**:

| Text | Correct Expression | Incorrect Expression |
|------|-------------------|---------------------|
| "over the weekend" | "weekend" | "over the weekend" |
| "during the summer" | "summer" | "during the summer" |
| "on Tuesday" | "Tuesday" | "on Tuesday" |

Note: The preposition may affect the interpretation (e.g., "over the weekend" implies the entire weekend), but the expression itself is typically just the temporal noun phrase.

## Handling Vague Duration Expressions

For vague expressions of duration or relative time, use the appropriate operators:

1. **For expressions implying a non-specific duration**:

| Expression | Correct Representation | Incorrect Representation |
|------------|------------------------|--------------------------|
| "within a few days" | `Next(interval=DCT, shift=Period(unit=DAY, n=None))` | `NextN(interval=DCT, shift=Period(DAY, 1), n=3)` |
| "in several weeks" | `Next(interval=DCT, shift=Period(unit=WEEK, n=None))` | `NextN(interval=DCT, shift=Period(WEEK, 1), n=X)` |
| "for some months" | `Next(interval=DCT, shift=Period(unit=MONTH, n=None))` | Specific number of months |

2. **Key points about vague durations**:
   - Use `n=None` to indicate an unspecified quantity
   - Prefer `Next` with a vague period over `NextN` with a specific count for naturally vague expressions
   - Don't try to quantify inherently vague expressions with specific numbers

3. **Common vague time patterns**:
   - "a few X" → Period(unit=X, n=None)
   - "several X" → Period(unit=X, n=None)

- "some X" → Period(unit=X, n=None)
- "many X" → Period(unit=X, n=None)

## ISO and Standard Date Format Handling

When annotating standard date formats, it's crucial to correctly identify the level of specificity intended in the expression:

1. **ISO format dates (YYYY-MM-DD)**:
   - Full date formats like "1998-08-07" should be annotated with day-level specificity, not as an entire year

| Expression | Correct Representation | Incorrect Representation |
|------------|------------------------|--------------------------|
| "1998-08-07" | `This(interval=Year(digits=1998), shift=RepeatingIntersection(shifts=[Repeating(unit=MONTH, range=YEAR, value=8), Repeating(unit=DAY, range=MONTH, value=7)]))` | `Year(digits=1998)` |
| "2020-12-31" | `This(interval=Year(digits=2020), shift=RepeatingIntersection(shifts=[Repeating(unit=MONTH, range=YEAR, value=12), Repeating(unit=DAY, range=MONTH, value=31)]))` | `Year(digits=2020)` |

2. **Common date format variants**:
   - American format (MM/DD/YYYY): "08/07/1998"
   - European format (DD/MM/YYYY): "07/08/1998"
   - All should be normalized to the same SCATE representation when they refer to the same date

3. **Metadata dates**:
   - Dates appearing in document headers, bylines, or metadata sections should be treated with the same precision they express
   - A standalone date like "1998-08-07" in a document header still refers to that specific day

4. **Key principle**: Always preserve the temporal granularity expressed in the original time expression. Never default to a broader time range (like a year) when a more specific one (like a day) is explicitly stated.

## Precise Annotation of Common Deictic Time Expressions

Common deictic time words require special attention to preserve their precise temporal meaning:

1. **Day-specific expressions**:

| Expression | Correct Representation | Incorrect Representation |
|------------|------------------------|--------------------------|
| "today" | `This(interval=DCT, shift=Repeating(unit=DAY, range=DAY))` | `This(interval=DCT, shift=None)` |
| "yesterday" | `Last(interval=DCT, shift=Repeating(unit=DAY, range=DAY))` | `Last(interval=DCT, shift=None)` |
| "tomorrow" | `Next(interval=DCT, shift=Repeating(unit=DAY, range=DAY))` | `Next(interval=DCT, shift=None)` |

2. **The importance of the `shift` parameter**:
   - Using `shift=None` loses the specific temporal granularity of the expression

- For words like "today", the `shift=Repeating(unit=DAY, range=DAY)` indicates it refers specifically to a day-long period
- This preserves the precise 24-hour meaning rather than treating it as a generic "current time" reference

3. **Other common deictic expressions and their correct representations**:

| Expression | Correct Representation |
|---|---|
| "this month" | `This(interval=DCT, shift=Repeating(unit=MONTH, range=MONTH))` |
| "this year" | `This(interval=DCT, shift=Repeating(unit=YEAR, range=YEAR))` |
| "next week" | `Next(interval=DCT, shift=Repeating(unit=WEEK, range=WEEK))` |
| "last night" | `Last(interval=DCT, shift=Night())` |

4. **Calendar-Aligned "last / past N" Expressions**:
   - Whenever the intent is to include **complete calendar units** (year, quarter, month, week, night, weekend …), choose `Repeating`, not `Period`.
   - `Period` is limited to **rolling** or **vague** durations that do **not** align exactly to named calendar units.

| Expression | Correct Representation | Why |
|---|---|---|
| "last year" | `Last(interval=DCT, shift=Repeating(unit=YEAR, range=YEAR))` | Use **Repeating** so that the interval aligns to **Jan 1 – Jan 1** of the previous calendar year, not a rolling 365-day window |
| "past three years" | `LastN(interval=DCT, shift=Repeating(unit=YEAR, range=YEAR), n=3)` | Same alignment logic as above |
| "last five quarters" | `LastN(interval=DCT, shift=Repeating(unit=QUARTER_YEAR, range=QUARTER_YEAR), n=5)` | Quarter-level repeat |

5. **Remember**: The `shift` parameter in temporal operators provides crucial information about the specific calendar unit being referenced, and should never be omitted for expressions with specific granularity.

## Core Time Expression Extraction and Modifier Separation

When identifying time expressions, it's crucial to separate core temporal references from their modifiers and determiners:

1. **Core time expressions vs. extended phrases**:

| Full Phrase | Core Time Expression | Modifiers (Not Part of Expression) |
| --- | --- | --- |
| "at the end of November" | "November" | "at the end of" |
| "today's editions" | "today" | "'s editions" |
| "by early December" | "December" | "by early" |
| "throughout next week" | "next week" | "throughout" |
| "in the past" | "past" | "in the" |

2. **Principles for identifying core time expressions**:
   - The core expression contains the essential temporal reference (day, month, year, etc.)
   - Adjectival markers like "'s" (possessive) usually indicate the end of the core expression
   - Prepositions ("at", "on", "by", "during") typically precede but aren't part of the core expression
   - Modifiers like "beginning of", "middle of", "end of" should be excluded from the core expression

3. **Special cases and exceptions**:
   - Certain temporal phrases form inseparable units: "end of year", "beginning of month"
   - Time-specific prepositions may be included when they change meaning: "in May" vs. "by May"
   - Possessive forms can semantically modify the time reference: "yesterday's" vs. "yesterday"
   - **Comparative-adverb + named-day phrases** (e.g., "earlier Sunday", "later Monday") must **not** be split;

4. **Correct annotation examples**:

| Text | Correct Annotation | Incorrect Annotation |
| --- | --- | --- |
| "The meeting is at the end of November" | time_text: "November" | time_text: "at the end of November" |
| "It was reported in today's newspaper" | time_text: "today" | time_text: "today's newspaper" |
| "They'll arrive by early next week" | time_text: "next week" | time_text: "by early next week" |

5. **Common error patterns to avoid**:
   - Including prepositions and qualifiers with the time expression
   - Including possessive markers and the modified nouns with the time expression
   - Splitting compound time expressions that should remain together (e.g., "next week", "last month")

# Articles and Quantifiers in Time Expressions

When annotating time expressions, it's important to understand how articles (a, an, the) and quantifiers interact with the core temporal expression:

1. **Articles in time expressions**:

| Full Text | Correct Time Expression | Notes |
|---|---|---|
| "a year earlier" | "year earlier" | The indefinite article "a" is not part of the time expression |
| "a week ago" | "week ago" | The indefinite article is excluded |
| "the previous month" | "previous month" | The definite article is excluded |

2. **Core time expression identification**:
   - Focus on the minimal span that conveys the complete temporal meaning
   - Articles ("a", "an", "the") typically fall outside the time expression boundary
   - Exception: When the article is integral to the meaning (e.g., "the day before yesterday")

3. **Relative time expressions with articles**:

| Expression | Correct Representation | Notes |
|---|---|---|
| "a year earlier" | `Before(interval=DCT, shift=Period(unit=YEAR, n=None))` | Use `n=None` for unspecified duration |
| "a year ago" | `Last(interval=DCT, shift=Period(unit=YEAR, n=1))` | Use specific `n=1` because "a year" specifies exactly one year |
| "a few weeks ago" | `Last(interval=DCT, shift=Period(unit=WEEK, n=None))` | Use `n=None` for vague durations like "a few" |

4. **Comparative time expressions**:
   - "earlier", "before", "prior", "previous" → Use `Before` operator
   - "later", "after", "following", "subsequent" → Use `After` operator
   - "ago" → Use `Last` operator

5. **Key points to remember**:
   - Focus on annotating just the minimum temporal expression needed to convey the complete meaning
   - Be consistent in excluding articles across all annotations
   - Pay attention to whether the time reference is specific (e.g., "a year" = exactly 1 year) or vague (e.g., "some years" = unspecified number of years)

## Example Annotations

When annotating time expressions, we need to output the time text and its corresponding SCATE representation. The output should be in JSON format with a list of objects, each containing:

- `time_text`: The text span containing the temporal expression
- `scate`: The SCATE code that represents this expression

Here's an example of the annotation process:

**Input:**

- DCT: `1998-05-01`
- Text: `The Internet, the global network of computers, is now far reaching into the country – extending its embrace to include every nook and cranny of the nation – opening doors to not only a diverse range of information sources but also an exhaustive list of possibilities to create new applications which add value to people's lives.`

**Output:**

```
[
  {
    "time_text": "now",
    "scate": "This(interval=Interval.of(1998, 5, 1), shift=None)"
  }
]
```

Here are more examples:

**Input:**

- DCT: `1998-05-01`
- Text: `APW19980501.0480\n\n05/01/1998 09:13:00\n\n\nAPW19980501.0480 NEWS STORY 05/01/1998 09:13:00\nw2844 Cx1f wstmr w Cx13 Cx11 SETTING-THE-STAGEsked 05-01 0665\nSETTING-THE-STAGE sked\nMALAYSIA Setting the stage for Net convergence, NEW STRAITS TIMES-MANAGEMENT TIMES QL xfdws SETTING-THE-STAGE sked Emerging Markets Datafile April 30, 1998 NEW STRAITS TIMES-MANAGEMENT TIMES ENGLISH COPYRIGHT 1998 BY WORLDSOURCES, INC., A JOINT VENTURE OF FDCH, INC.\nAND WORLD TIMES, INC.\nNO PORTION OF THE MATERIALS CONTAINED HEREIN MAY BE USED IN ANY MEDIA WITHOUT ATTRIBUTION TO WORLDSOURCES, INC.\n\n\n\nMALAYSIA's aggressive move into the information age could not come in a more opportune time.`

**Output:**

```
[
    {
    "time_text": "05/01/1998 09:13:00",
    "scate": "This(interval=Year(digits=1998), shift=RepeatingIntersection(shifts=
[Repeating(unit=MONTH, range=YEAR, value=5), Repeating(unit=DAY, range=MONTH, value=1),
Repeating(unit=HOUR, range=DAY, value=9), Repeating(unit=MINUTE, range=HOUR, value=13),
Repeating(unit=SECOND, range=MINUTE, value=0)]))"
  }
```

```
  {
    "time_text": "1998",
    "scate": "Year(digits=1998)"
  },
   {
    "time_text": "April 30, 1998",
    "scate": "This(interval=Year(digits=1998), shift=RepeatingIntersection(shifts=
[Repeating(unit=MONTH, range=YEAR, value=4), Repeating(unit=DAY, range=MONTH,
value=30)]))"
  },
  {
    "time_text": "time",
    "scate": "This(interval=Interval.of(1998, 5, 1), shift=None)"
  },
]
```

## More Important Notes for Annotation

1. The document creation time (DCT) is provided as input along with the text

2. Focus on identifying all time expressions in the text and writing the correct SCATE code for each

3. Always output in JSON format with a list of objects containing `time_text` and `scate` fields

4. Use the appropriate SCATE operators and classes based on the time expression semantics

5. For deictic expressions (like "now", "today", "yesterday"), use the DCT as the reference point

6. For relative expressions (like "next Friday", "last month"), anchor them to the DCT A collection of intervals on the timeline. This class implements the Python Iterable interface to allow iteration over a set of Interval objects.

7. Time expressions in the output JSON must be ordered according to their appearance in the original text

8. Always consider the hierarchical nature of time expressions and their composition

9. Pay special attention to temporal adverbs and prepositions:
   - Words indicating past time (`previously`, `before`, `earlier`, `formerly`, `in the past`, etc.) should typically use the `Last` operator, not `This`
   - Words indicating future time (`soon`, `later`, `in the future`, etc.) should typically use the `Next` operator, not `This`
   - Words indicating present time (`currently`, `presently`, `at the moment`, etc.) should use the `This` operator

## Common Annotation Pitfalls

1. **Misinterpreting vague temporal adverbs**: Words like "previously" or "formerly" indicate a time before the reference time, so they should be annotated with `Last(interval=DCT, shift=None)` rather than `This(interval=DCT, shift=None)`.

2. **Confusing This vs. Last**: The `This` operator refers to the current time interval, while `Last` refers to a previous time interval. For example:

```
# Incorrect (for "previously")
This(interval=Interval.of(1989, 11, 2), shift=None)

# Correct (for "previously")
Last(interval=Interval.of(1989, 11, 2), shift=None)
```

3. **Ignoring contextual cues**: Sometimes temporal expressions must be interpreted based on surrounding context in the text, not just the DCT.

4. **Incorrect time expression boundary detection**: Be careful when identifying the boundaries of time expressions. Pay attention to:
   - Time expressions that span multiple tokens (e.g., `08-07-98 0618` should be treated as a single expression)
   - Time zone indicators that are not part of the expression (e.g., EDT, GMT, UTC)
   - Whitespace that may separate parts of a single time expression

5. **Fragmentation of combined date-time expressions**: Date and time components that appear together (e.g., "08-07-98 0618") should be treated as a single time expression, not split into separate expressions.

6. **Missing implicit time references**: Some texts imply a time reference without explicitly stating it (e.g., "previously" without stating when).

# C   Interval Few-shot Prompt

**Input:**

- DCT: 1998-05-01

- Text: The Internet, the global network of computers, is now far reaching into the country - extending its embrace to include every nook and cranny of the nation - opening doors to not only a diverse range of information sources but also an exhaustive list of possibilities to create new applications which add value to people's lives.

**Output:**

```
[
  {
    "time_text": "now",
    "normalized": "... ..."
  }
]
```

**Input:**

- DCT: 1998-05-01

- Text: APW19980501.0480\n\n05/01/1998 09:13:00\n\n\nAPW19980501.0480 NEWS STORY 05/01/1998 09:13:00\nw2844 Cx1f wstmr w Cx13 Cx11 SETTING-THE-STAGEsked 05-01 0665\nSETTING-THE-STAGE sked\nMALAYSIA Setting the stage for Net convergence, NEW STRAITS TIMES-MANAGEMENT TIMES QL xfdws SETTING-THE-STAGE sked Emerging Markets Datafile April 30, 1998 NEW STRAITS TIMES-MANAGEMENT TIMES ENGLISH COPYRIGHT 1998 BY WORLDSOURCES, INC., A JOINT VENTURE OF FDCH, INC.\nAND WORLD TIMES, INC.\nNO PORTION OF THE MATERIALS CONTAINED HEREIN MAY BE USED IN ANY MEDIA WITHOUT ATTRIBUTION TO WORLDSOURCES, INC.\n\n\nMALAYSIA's aggressive move into the information age could not come in a more opportune time.

**Output:**

```
[
    {
    "time_text": "05/01/1998 09:13:00",
    "scate": "1998-05-01T09:13:00 1998-05-01T09:13:01"
  }
  {
    "time_text": "1998",
    "scate": "1998-01-01T00:00:00 1999-01-01T00:00:00"
  },
    {
    "time_text": "April 30, 1998",
    "scate": "This(interval=Year(digits=1998), shift=RepeatingIntersection(shifts=
[Repeating(unit=MONTH, range=YEAR, value=4), Repeating(unit=DAY, range=MONTH,
value=30)]))"
  },
  {
    "time_text": "time",
    "scate": "1998-04-30T00:00:00 1998-05-01T00:00:00"
```

```
    },
  ]
```

**Input:**

- DCT: 1989-11-02
- Text: Integra-A Rights Offering 11/02/89 WALL STREET JOURNAL (J) ITGR HWG IRVING, Texas\n\n\n\nIntegra-A Hotel amp Restaurant Co. said its planned rights offering to raise about $9 million was declared effective and the company will begin mailing materials to shareholders at the end of this week.

**Output:**

```
[
  {
    "time_text": "this week",
    "normalized": "1989-10-30T00:00:00 1989-11-06T00:00:00"
  },
  {
    "time_text": "11/02/89",
    "normalized": "1989-11-02T00:00:00 1989-11-03T00:00:00"
  }
]
```

**Input:**

- DCT: 1989-11-02
- Text: Stephen Akerfeldt, currently vice president finance, will succeed Mr. McAlpine.

**Output:**

```
[
  {
    "time_text": "currently",
    "normalized": "... ..."
  }
]
```

# D  Additional Comparisons and Statistical Significance

## D.1  Comparison with Existing Systems

We compare our method against both SCATE-based and ISO-TimeML-based time normalization systems.

**SCATE-based baseline**    We compare against the Neural Parser by Xu et al. [2019], which represents the only publicly available SCATE-based implementation and the previous state-of-the-art. On the test set, the Neural Parser achieves an F1 score of 0.43 (Precision: 0.57, Recall: 0.35). Our method improves upon this by 37% in F1 score (from 0.43 to 0.59), demonstrating the effectiveness of our code generation approach over the previous multi-stage LSTM pipeline.

**ISO-TimeML baseline**    We also compare against SUTime [Chang and Manning, 2012], a widely-used rule-based time normalization system. SUTime achieves an F1 score of 0.45 (Precision: 0.58, Recall: 0.37) on our test set, while our method achieves 0.59, demonstrating the advantages of the more expressive SCATE framework.

**Runtime comparison**    We conduct runtime measurements on the test set using an NVIDIA RTX 3090 GPU with vLLM [Kwon et al., 2023] for efficient inference. Our Qwen2.5-0.5B model processes the entire test set in 6 seconds, compared to 190 seconds for the Neural Parser's LSTM-based pipeline, achieving a 31.7× speedup. This significant improvement stems from: (1) end-to-end generation eliminating multi-stage pipeline overhead, (2) efficient transformer-based inference, and (3) smaller model size (0.5B parameters) enabling faster forward passes.

## D.2  Confidence Intervals

We calculate 95% confidence intervals using bootstrap sampling, where we randomly resample 80% of the evaluation data 100 times and compute evaluation metrics on each sample to estimate the statistical uncertainty of our model's performance. We present the results in Table S1 and Table S2.

Table S1: Performance of LLMs on Temporal Expression Recognition and SCATE Code Generation on Training Set with 95% confidence intervals. Values shown as mean ± std, confidence intervals (CI): (lower, upper).

| Model | Accuracy | Precision | Recall | F1 |
|---|---|---|---|---|
| Claude 3.5 | 0.62 ± 0.02 CI: (0.62, 0.63) | 0.64 ± 0.02 CI: (0.64, 0.65) | 0.62 ± 0.02 CI: (0.62, 0.63) | 0.63 ± 0.02 CI: (0.63, 0.64) |
| Claude 3.7 | **0.69 ± 0.02 CI: (0.69, 0.70)** | 0.63 ± 0.02 CI: (0.63, 0.64) | **0.69 ± 0.02 CI: (0.69, 0.70)** | **0.66 ± 0.02 CI: (0.66, 0.67)** |
| Gemini 2.0 Flash | 0.65 ± 0.03 CI: (0.64, 0.65) | 0.64 ± 0.02 CI: (0.63, 0.64) | 0.65 ± 0.03 CI: (0.64, 0.65) | 0.64 ± 0.03 CI: (0.64, 0.65) |
| Gemini 2.5 Flash | 0.61 ± 0.02 CI: (0.61, 0.62) | 0.63 ± 0.02 CI: (0.63, 0.64) | 0.61 ± 0.02 CI: (0.61, 0.62) | 0.62 ± 0.02 CI: (0.62, 0.63) |
| Gemini 2.5 Pro | 0.50 ± 0.02 CI: (0.50, 0.51) | **0.65 ± 0.02 CI: (0.64, 0.65)** | 0.50 ± 0.02 CI: (0.50, 0.51) | 0.56 ± 0.02 CI: (0.56, 0.57) |
| GPT-4.1 | 0.66 ± 0.02 CI: (0.66, 0.67) | 0.60 ± 0.02 CI: (0.60, 0.60) | 0.66 ± 0.02 CI: (0.66, 0.67) | 0.63 ± 0.02 CI: (0.63, 0.63) |

# E  License information

We respect the license and intended use of all models and datasets employed in this study. Detailed license information is provided below.

Table S2: Performance comparison of different methods on the test set with 95% confidence intervals. Values shown as mean ± std, confidence intervals (CI): (lower, upper).

| Methods | Accuracy | Precision | Recall | F1 |
|---|---|---|---|---|
| Qwen2.5-0.5B + Training Set | 0.01 ± 0.01 CI: (0.00, 0.01) | 0.49 ± 0.50 CI: (0.39, 0.59) | 0.01 ± 0.01 CI: (0.00, 0.01) | 0.01 ± 0.01 CI: (0.01, 0.01) |
| Qwen2.5-0.5B + CC-News | 0.37 ± 0.03 CI: (0.36, 0.37) | 0.46 ± 0.04 CI: (0.45, 0.46) | 0.37 ± 0.03 CI: (0.36, 0.37) | 0.41 ± 0.03 CI: (0.40, 0.41) |
| Qwen2.5-0.5B + CC-News + Training Set | **0.59 ± 0.04 CI: (0.58, 0.60)** | **0.59 ± 0.04 CI: (0.59, 0.60)** | **0.59 ± 0.04 CI: (0.58, 0.60)** | **0.59 ± 0.04 CI: (0.58, 0.60)** |
| Claude 3.7 + SCATE Prompt | 0.49 ± 0.03 CI: (0.49, 0.50) | 0.56 ± 0.04 CI: (0.55, 0.57) | 0.49 ± 0.03 CI: (0.49, 0.50) | 0.52 ± 0.03 CI: (0.52, 0.53) |
| Claude 3.7 + Interval Few-shot Prompt | 0.38 ± 0.03 CI: (0.38, 0.39) | 0.39 ± 0.03 CI: (0.38, 0.39) | 0.38 ± 0.03 CI: (0.38, 0.39) | 0.38 ± 0.03 CI: (0.38, 0.39) |
| GPT 4.1 + SCATE Prompt | 0.51 ± 0.03 CI: (0.51, 0.52) | 0.51 ± 0.03 CI: (0.50, 0.52) | 0.51 ± 0.03 CI: (0.51, 0.52) | 0.51 ± 0.03 CI: (0.50, 0.52) |

**Models.** The Claude family models utilized in our study are licensed under the Commercial Terms of Service. The Gemini family models are licensed under the Google APIs Terms of Service. The GPT-4.1 model is licensed under the Business terms. The Qwen 2.5 models are licensed under the Apache License 2.0.

**Datasets.** The CC-News dataset used in our study is available under the Common Crawl Terms of Use.

## F  Detailed Error Analysis

To better understand the limitations of our best-performing model (Qwen2.5-0.5B + CC-News + Training Set), we conduct a detailed error analysis on the test set predictions. We categorize incorrect predictions into five main types based on their root causes and provide representative examples for each category.

### F.1  Error Distribution

We identify the following error distribution:

- **Missed expressions** (70%): The model fails to identify temporal expressions that are annotated in the gold standard.
- **Boundary errors** (10%): The model identifies a temporal expression but with incorrect span boundaries.
- **Structural errors** (10%): The model fails to capture the correct compositional structure of complex temporal expressions.
- **Operator confusion** (5%): The model selects an inappropriate SCATE operator despite correct span identification.
- **Granularity errors** (5%): The model represents temporal expressions at incorrect levels of granularity.

### F.2  Missed Expressions

The most critical failure mode occurs when the model completely fails to identify annotated temporal expressions. These account for 70% of all errors.

**Example**

- **Context**: "We estimated we could do it in 100 days, and we got across on the 99th day."
- **Gold expression**: "99th day"

- **Gold SCATE**: `Nth(shift=Repeating(unit=DAY), index=99)`
- **Predicted**: Missing
- **Analysis**: The model struggles with ordinal expressions embedded in narrative contexts, possibly because they are less explicit than standard date formats.

## F.3 Boundary Errors

Boundary errors occur when the model identifies the general location of a temporal expression but fails to capture the exact span. These represent 10% of errors.

**Example**

- **Context**: "He said: 'Lowe was a brilliant, kind fellow who never sought the limelight... and 60 years on from Everest his achievements deserve wider recognition.'"
- **Gold annotation**: "60 years on from"
- **Model prediction**: "60 years on" (missing "from")
- **Analysis**: Both generate similar SCATE code, but the boundary difference affects evaluation. This suggests the model understands the temporal semantics but struggles with precise span detection, particularly for multi-word prepositions.

## F.4 Structural Errors

Structural errors involve complex nested temporal expressions requiring compositional operators. These account for 10% of errors.

**Example**

- **Context**: "The season started about a month earlier than usual, sparking concerns it might turn into the worst in a decade"
- **Gold expression**: "month earlier than usual, sparking concerns it might turn into the worst in a decade"
- **Gold SCATE**: `Last(interval=Before(interval=Interval(...), shift=Period(unit=MONTH)), shift=Period(unit=DECADE))`
- **Predicted**: Only captures fragments ("a month earlier", "decade") separately
- **Analysis**: The model cannot recognize that these fragments form a single complex comparative temporal expression requiring nested operators.

## F.5 Operator Confusion

Operator confusion occurs when the span is correctly identified but the wrong SCATE operator is selected. These represent 5% of errors.

**Example**

- **Context**: "Mr. Obama said later at a news conference in Amman..."
- **Gold expression**: "later"
- **Gold SCATE**: `After(interval=Interval(...), shift=None)`
- **Predicted SCATE**: `Next(interval=Interval(...), shift=None)`
- **Analysis**: Both `After` and `Next` operators indicate future time, but `After` is more appropriate for indefinite future references while `Next` implies the immediate next occurrence. This reflects the subtle semantic distinctions between SCATE operators.

## F.6 Granularity Errors

Granularity errors occur when the model represents temporal expressions at incorrect levels of detail. These account for 5% of errors.

**Example**

- **Context**: "One exception was the swine flu pandemic of 2009-2010, when 348 children died."
- **Gold expression**: "2009-2010"
- **Gold SCATE**: `Between(start_interval=Year(digits=2009), end_interval=Year(digits=2010), start_included=True, end_included=True)`
- **Predicted SCATE**: `Year(digits=2009)`
- **Analysis**: The model loses the year range information and represents only the first year. The gold annotation requires both 2009 and 2010 to be fully included (spanning to 2011-01-01), but the model only captures 2009, demonstrating a granularity error where critical temporal scope is lost.

## F.7 Implications and Future Directions

Our error analysis reveals that the primary challenge lies in span recognition rather than SCATE code generation. Once a temporal expression is correctly identified, the model generally produces appropriate SCATE code. The high proportion of missed expressions (70%) suggests that improving temporal expression recognition represents the most promising direction for future work, either through better training data, architectural improvements, or specialized span detection modules. The remaining 30% of errors related to boundaries, structure, and operator selection could potentially be addressed through more comprehensive training examples covering edge cases and rare compositional patterns.

