# OpenReview forum: "A Semantic Parsing Framework for End-to-End Time Normalization"
_NeurIPS.cc/2025/Conference — NeurIPS 2025 poster_

### Official Review · Reviewer_FPoj · 2025-06-24

**Clarity:** 4
**Significance:** 3
**Originality:** 3
**Rating:** 4
**Confidence:** 3

**Summary:**

This paper proposes a novel end-to-end approach to time normalization by casting it as a code generation task within the SCATE (Semantically Compositional Annotation of Temporal Expressions) framerwork. The authors implement SCATE as a python library and show that large language models can generate valid code directly from natural language. They introduce a data augmentation pipeline in which LLMs generate annotated examples that are automatically validated through code execution. Using this synthetic data, they fine-tune a smaller, local model and report that it outperforms the LLMs originally used to generate the training data. Overall, the work presents a scalable, interpretable, and efficient method for handling complex temporal expressions.

The authors propose an innovative data augmentation strategy where LLMs generate synthetic annotated data (sentence, temporal expression, SCATE code) that is validated through runtime execution of the code, ensuring correctness. They demonstrate that a small, locally deployable model (Qwen2.5-0.5B) fine-tuned on this augmented dataset can outperform the LLMs used for data generation, achieving competitive performance while being computationally efficient and interpretable.

**Questions:**

Have you considered decoupling span detection from code generation via a two-stage pipeline? Would introducing a lightweight span classifier before code generation improve performance and reduce missed expressions?

Does the framework work on non-english languages?

Could your executable code generation approach extend to other semantic parsing or reasoning tasks such as event extraction?

Any failure analysis on the discarded code samples that failed execution?

Any biases from the LLM generated dataset?

**Ethical Concerns:**

["NO or VERY MINOR ethics concerns only"]

**Final Justification:**

The paper is a borderline accept due to the data limitations and the lack of experiments in an expanded the domain.

**Limitations:**

The limitations in the paper are well-articulated, including:

Restriction to TempEval-2013 for SCATE evaluation.

Lack of multilingual or open-domain benchmarks.

Heavy reliance on proprietary LLMs for data generation.

Performance gaps in span detection.

Pending open-source release of code and datasets.

**Paper Formatting Concerns:**

None that I could see

**Quality:**

3

**Strengths And Weaknesses:**

Strengths

1. Methodological Innovation:
The formulation of time normalization as a code generation problem using executable SCATE operators is novel. By directly outputting executable code, the system offers clear interpretability and determinism, unlike black-box sequence-to-sequence models.

2. Practical Implementation:
The authors fully implement SCATE as a Python library with composable, object-oriented design, making the semantics both human-readable and machine-executable. The paper carefully details the class structures for intervals, repeating intervals, periods, and operators, ensuring transparency.

3. LLM-driven Data Augmentation with Validation:
The automatic data augmentation pipeline cleverly uses LLMs to generate large-scale annotated data, filtered via code execution to guarantee syntactic and semantic correctness. This reduces reliance on scarce expert-annotated datasets and improves scalability.

Weaknesses:
Dataset and Domain limitations:

The approach is evaluated solely on the TempEval-2013 dataset. The paper does not address how the system performs in other domains or languages beyond english.

The system struggles with accurately identifying temporal expression spans.

Public release not available yet

Paper does not explore open source LLM alternatives, focuses mainly on Claude and GPT-4.1

---

> ### Author Rebuttal · Authors · 2025-07-31
>
> Thank you for your positive and constructive review. We greatly appreciate your recognition of our methodological innovation, practical implementation, and LLM-driven data augmentation approach. We address your concerns below:
>
> **W1: Dataset and domain limitations**
>
> We acknowledge the limitation to TempEval-2013. Unfortunately, this is the only publicly available dataset with SCATE annotations. While medical domain SCATE annotations exist, privacy restrictions prevent access to the underlying text.
>
> However, our data augmentation method is domain-agnostic and can be applied to generate annotations for any domain. In fact, we successfully augmented news data (CC-News) in our experiments. Extending to more domains through automatic annotation is a key direction for future work.
>
> **W2: Span detection struggles**
>
> While span detection remains challenging, we emphasize that our method achieves state-of-the-art performance on this difficult task. During the rebuttal period, we conducted additional comparisons:
>
> Existing baselines:
> - Neural Parser (Xu et al., 2019) - The only publicly available state-of-the-art SCATE system before our work: F1: 0.4317
> - SUTime (ISO-TimeML-based): F1: 0.4510
>
> Our method achieves F1: 0.59 - a 36.7% improvement over the previous SCATE state-of-the-art and 30.8% over SUTime.
>
> The span detection challenge is inherent to the complexity of temporal expressions, particularly:
> - Compositional expressions like "the past three summers" where understanding the full semantic scope requires compositional reasoning
> - Context-dependent boundaries where the same phrase may or may not be temporal (e.g., "spring" as season vs. mechanical spring)
> - Multi-span expressions that SCATE uniquely handles but are impossible in simpler frameworks
>
> Our end-to-end approach is the first to achieve robust performance on these complex cases without requiring separate models for recognition, linking, and normalization. While there's room for improvement, we've established a new state-of-the-art baseline. Further improving span detection through larger-scale data generation and better modeling of compositional semantics is a promising direction for future work.
>
> We will add these comparisons and analysis to the revised manuscript.
>
> **W3: Public release**
>
> We are fully committed to releasing all code, models, and datasets upon publication. This includes:
> - Complete SCATE Python library implementation
> - Fine-tuned Qwen2.5-0.5B model
> - 8,583 augmented training examples
> - Training and evaluation scripts
>
> **W4: Open-source LLM alternatives**
>
> We would like to clarify that training and releasing an open-source small model (Qwen2.5-0.5B) is one of our main contributions. For data augmentation, we did explore open-source LLMs from the Llama family, but their performance was significantly below GPT-4.1 and Claude. However, our method is model-agnostic and will benefit from improvements in open-source LLMs.
>
> ### Questions:
>
> **Q1: Two-stage pipeline for span detection**
>
> Our end-to-end approach is motivated by both practical and technical considerations:
>
> Practical motivation: Time normalization is often an upstream task in large-scale applications. For example, when processing millions of documents to build temporal knowledge graphs, a two-stage pipeline significantly increases complexity and reduces efficiency. Single-model inference is crucial for production deployments.
> Experimental validation: During the rebuttal period, we tested a two-stage approach using the CC-News + Training Set as training set:
>
> - Stage 1: Train BERT-base for time expression extraction
> - Stage 2: Train Qwen for SCATE code generation
> - Results: F1: 0.2490 (vs. 0.59 for our joint model)
> - Critical failure: 251/313 expressions missed (80% miss rate)
>
> While we acknowledge this BERT model here wasn't heavily tuned and more NER-specific data could help, the fundamental issues remain:
>
> - Error propagation: Missed expressions cannot be recovered in stage 2
> - Semantic coupling: Expression boundaries often depend on their SCATE interpretation (e.g., "the past three summers" - the full phrase is needed to understand it's a time expression), while separate extraction may miss context
> - Deployment complexity: Maintaining two models with different architectures
>
> Our joint approach leverages SCATE semantics during extraction, leading to better recognition and normalization. We agree specialized extraction techniques are promising for future work, particularly continued large-scale data generation to improve end-to-end models in this area. We will add these experiments and discussion to the manuscript.
>
> **Q2: Non-English languages**
>
> Yes, the framework is language-agnostic. As long as SCATE annotations are designed for the target language, our approach can be applied. The Python library and code generation method work independently of the input language.
>
> **Q3: Extension to other tasks**
>
> As demonstrated in an exsiting paper "A Dataset and Evaluation Framework for Complex Geographical Description Parsing", similar operators (Union, Intersection) apply to geo-normalization tasks. Our executable code generation approach can be readily extended to any semantic parsing task with compositional semantics, including event extraction, spatial reasoning, and logical form generation.
>
> **Q4: Failure analysis on discarded samples**
>
> Discarded samples primarily failed due to:
>
> Syntax errors:
> - Missing parentheses: `Year(2023` instead of `Year(2023)`
> - Undefined variables: Using `Months` instead of `MONTH` enum
> - Invalid Python syntax: Malformed string literals or incorrect indentation
>
> Semantic errors:
> - Invalid operator combinations: `Next(shift=Period(...))` when Next requires a Repeating shift
> - Incorrect parameter types: `Repeating(unit=DAY, value="Monday")` instead of numeric value
> - Incompatible nesting: `This(interval=Period(...))` when This requires an Interval
>
> Runtime errors:
> - None value operations: Attempting arithmetic on underspecified expressions
> - Invalid date operations: `Interval.of(2023, 14, 1)` with month 14
> - Infinite loops in repeating intervals without proper bounds
>
> These failures demonstrate our validation mechanism's effectiveness: only code that is both syntactically correct and semantically meaningful passes through, ensuring high-quality training data. We will include this analysis in the revised manuscript.
>
>
> **Q5: Biases in LLM-generated data**
>
> We tested our augmented dataset using the `vector-institute/Llama3.2-NLP-Newsmedia-Bias-Detector` model. The analysis found no biases in the generated temporal expressions.
>
> Thank you again for your constructive feedback. We will incorporate these clarifications and experimental results in our revision.

---

> > ### Comment · Reviewer_FPoj · 2025-07-31
> >
> > Thanks for your response. While I believe you answered most of my questions. I feel the paper is still a borderline accept due to the data limitations and the lack of experiments in an expanded the domain.

---

### Official Review · Reviewer_YsgP · 2025-07-01

**Clarity:** 3
**Significance:** 2
**Originality:** 3
**Rating:** 4
**Confidence:** 3

**Summary:**

This paper presents a novel approach to time normalization by reformulating it as a code generation problem following the SCATE framework. Traditional time normalization approaches that are based on ISO-TimeML schema struggle with complex temporal expressions like event-relative or compositional phrases. The presented method addresses this by implementing a fully executable SCATE Python library that translates temporal expressions expressed as text into structured and interpretable Python code, which then gets executed to produce deterministic normalized time normalization.

Contributions of this paper include:
- It implements SCATE as a Python library, enabling deterministic execution of normalized time intervals;
- It proposes data augmentation by using LLMs to generate SCATE code from text following a well-crafted prompt, creating over 8,000 new annotations of text-code pairs;
- It presents empirical results of fine-tuning a small-scale model (Qwen2.5-0.5B) on the augmented data, showing that such a model can outperform the LLMs used for data generation and providing further analysis.

**Questions:**

As I mentioned, a main weakness of this paper is that the experiment section lacks thoroughness. My questions below are mainly around a result of that:
- How does the proposed method compare to previous methods in a similar evaluation setting?

- The authors show that in order for the 0.5B to learn the task, the augmented data (8k+) has to be used with the training set (300 examples), or otherwise the performance is close to zero. This suggests some data efficiency issues that can be addressed with options below. Have these been explored?
  * efficient fine-tuning methods (LoRA), which typically works better with less data;
  * using code-finetuned checkpoint instead of general instruction-tuned model.

**Ethical Concerns:**

["NO or VERY MINOR ethics concerns only"]

**Final Justification:**

After reading the authors' response, I have adjusted my overall score from 3 to 4. As mentioned in my original review, the original version of the manuscript was weak because of its brief experiments, lack of insightful analysis, and relatively niche focus compared to other papers with broader audience and impact. The authors' response have improved the analysis part of the paper, and partially improved the thoroughness of its experiments. My concern on its niche focus remains and I do still think the experiments can be made even more extensive with more comparisons, therefore I am giving an overall rating of 4 (borderline accept).

**Limitations:**

yes

**Quality:**

3

**Strengths And Weaknesses:**

Strengths:
- The method is original and has novelty according to the related work discussion. The idea of formulating time normalization as a code generation problem makes a lot of sense (i.e., the proposed python implementation essentially serves as a DSL for the time normalization task). This said, I am not very familiar with related work so cannot judge with confidence if the authors have missed important related work.
- The paper is presented well and is very easy to follow and understand.
- The empirical results are overall positive, showing that the approach is promising. The derived data as well as the model can be a useful resource for the community.

Weaknesses:
- The biggest weakness of this work is that the experiment section is brief and not particularly thorough, leaving some questions unanswered. As an example, it is hard to know how this method compares to previous work in a similar evaluation setup. I'll include more thoughts in the questions section below.
- The problem being studied is relatively niche and is only impactful to a small community, limiting its overall significance.
- Some in-depth analysis with examples can be provided to help the readers better understand the behaviors and limitations of the model.

---

> ### Author Rebuttal · Authors · 2025-07-31
>
> Thank you for your review and constructive feedback. We appreciate your recognition of our novel approach and the potential value of our resources to the community. We address your concerns below:
>
> **W1: Limited experimental evaluation and comparison with previous work**
> During the rebuttal period, we conducted additional comparisons.
>
> Existing SCATE-based system:
> - Neural Parser (Xu et al., 2019) - The only publicly available SCATE implementation before our work: F1: 0.4317, Precision: 0.5677, Recall: 0.3482
>
> - SUTime (ISO-TimeML-based system): F1: 0.4510, Precision: 0.5838, Recall: 0.3674
>
> Our method:
>
> - Qwen2.5-0.5B + CC-News + Training Set: F1: 0.59, Precision: 0.59, Recall: 0.59
> - 36.7% improvement over the previous SCATE state-of-the-art
> - 30.8% improvement over SUTime
>
> These results demonstrate that our approach significantly outperforms existing methods in comparable evaluation settings. We will add these comparisons to the revised manuscript.
>
> **W2: Data efficiency issues**
>
> During the development, we explored several strategies:
>
> 1. Different fine-tuning approaches: We experimented with LoRA during development. However, we observed that LoRA produced similar results to full fine-tuning, particularly because our model is already quite small (0.5B parameters). Given the minimal difference and the simplicity of full fine-tuning for such a small model, we opted for full fine-tuning in our final experiments.
> 2. Model selection rationale: We also explored code-pretrained models during development. However, we found that small code-specific models like Qwen2.5-Coder struggled significantly with temporal expression identification, leading to numerous missed time expressions before even attempting SCATE generation. This resulted in severe performance degradation (more than 0.2 F1 score drop). The issue appears to be that code-pretrained models, while excellent at code generation, lack the natural language understanding capabilities needed for identifying temporal expressions in text. Since our task requires both strong NLP capabilities (for span identification) and code generation abilities, we found that general instruction-tuned models like Qwen2.5-0.5B-Instruct provide the best balance. We acknowledge that exploring larger code-specific models or hybrid approaches could be valuable future work, but for our goal of deployable small models, the general instruction-tuned model proved most effective.
>
>
> **W3: In-depth analysis and examples**
>
> We conducted detailed error analysis on our best model's predictions. Here are specific examples illustrating the main failure modes:
>
> 1. Missed expressions (70% of errors): These are the most critical failures where the model completely fails to identify annotated time expressions:
>
> - Example: "We estimated we could do it in 100 days, and we got across on the 99th day."
>  - Time expression: "99th day"
>  - Gold: `Nth(shift=Repeating(unit=DAY), index=99)`
>  - Predicted: Missing
>  - Analysis: The model struggles with ordinal expressions embedded in narrative contexts.
>
> - Example: "This flu season started in early December..."
>  - Time expression: "This flu season"
>  - Gold: `This(interval=Interval.of(2013, 3, 22), shift=Repeating(unit=None))`
>  - Predicted: Missing
>  - Analysis: Abstract temporal concepts like "season" without explicit calendar anchors are challenging.
>
> 2. Boundary errors (10% of errors): The model identifies temporal expressions but with incorrect boundaries:
>
> - Example: "He said: 'Lowe was a brilliant, kind fellow who never sought the limelight... and 60 years on from Everest his achievements deserve wider recognition.'"
>  - Gold annotation: "60 years on from"
>  - Model prediction: "60 years on" (missing "from")
>  - Both generate similar SCATE code but the boundary difference affects evaluation
>  - This suggests the model understands the temporal semantics but struggles with precise span detection
>
> 3. Structural errors (10% of errors): Complex nested temporal expressions requiring compositional operators:
>
> - Example: "The season started about a month earlier than usual, sparking concerns it might turn into the worst in a decade"
>  - Time expression: "month earlier than usual, sparking concerns it might turn into the worst in a decade"
>  - Gold: `Last(interval=Before(interval=Interval(...), shift=Period(unit=MONTH)), shift=Period(unit=DECADE))`
>  - Predicted: Only captures fragments ("a month earlier", "decade") separately
>  - Analysis: The model cannot recognize that these fragments form a single complex comparative temporal expression
>
> 4. Operator confusion (5% of errors): Correct span identification but wrong SCATE operator selection:
>
> - Example: "Mr. Obama said later at a news conference in Amman..."
>  - Time expression: "later"
>  - Gold: `After(interval=Interval(...), shift=None)`
>  - Predicted: `Next(interval=Interval(...), shift=None)`
>  - Analysis: Both operators indicate future time, but "After" is more appropriate for indefinite future references while "Next" implies the immediate next occurrence
>
> 5. Granularity errors (5% of errors):
>
> - Example: "Mr Lowe also took part in the trans-Antarctic expedition of 1957-58"
>  - Time expression: "1957-58"
>  - Gold: `Between(start_interval=Year(1957), end_interval=YearSuffix(interval=Year(1957), digits=58))` → spans 1957-1959
>  - Predicted: `Between(start_interval=Year(1957), end_interval=Year(1958))` → spans only 1958
>  - Analysis: The model misinterprets abbreviated year ranges, treating "58" as 1958 rather than understanding the convention that "1957-58" means 1957-1958 inclusive
>
> We will add this detailed analysis with examples to the revised manuscript to help readers better understand the model's capabilities and limitations.
>
> **Additional clarifications on significance of the problem**
>
> While time normalization may appear to be a niche problem, it is actually a fundamental building block that quietly powers many critical real-world systems:
>
> - Clinical decision support: Accurate interpretation of expressions like "take medication three times daily for two weeks" or "symptoms started 3 days post-surgery" can be life-critical
> - Legal document analysis: Misinterpreting "within 30 days of receipt" vs "30 days after receipt" in contracts can have million-dollar consequences
> - Financial systems: Trading algorithms and regulatory compliance depend on precise temporal reasoning for transaction windows and reporting deadlines
> - Question answering: Any system answering "when" questions requires robust time normalization as a core component
>
> The challenge is that these applications often fail silently when time normalization is incorrect: a QA system simply returns wrong dates, a clinical system miscalculates medication schedules. Our work provides the first publicly available, executable implementation of the most expressive temporal framework (SCATE), along with a large-scale dataset and deployable model. This enables researchers and practitioners to address complex temporal expressions that were previously impossible to handle systematically. In essence, time normalization is like parsing in NLP: a seemingly narrow technical problem that underlies countless applications. Our contribution makes this critical component more accessible and capable.
>
> Thank you again for your constructive feedback. We will incorporate these clarifications and experimental results in our revision.

---

> > ### Comment · Reviewer_YsgP · 2025-08-07
> >
> > I want to thank the authors for their detailed response. I have adjusted my ratings accordingly.
> >
> > The response has addressed my concern on the lack of good, in-depth analysis. It is good to see the error analysis and the conclusions draw from it. The manuscript will be improved significantly with the analysis added.
> >
> > The additional experiments on previous systems are helpful too - they provide good baselines to better judge the significance of the method. Ideally the manuscript can be further enriched with detailed experiments on the "data efficiency" part, but adding the current text into discussions and analysis will certainly help too.
> >
> > I am still not entirely convinced by the argument on the broader impact. My concern of its relatively niche focus remains after reading the authors' response.

---

### Official Review · Reviewer_wavY · 2025-07-02

**Clarity:** 4
**Significance:** 3
**Originality:** 3
**Rating:** 5
**Confidence:** 3

**Summary:**

The authors present a novel formulation of the time normalization task as a code generation task. They contribute a comprehensive python library for executing the existing SCATE framework, which represents temporal expressions using temporal operators. They then investigate the use of large language models to generate SCATE code within their novel python library, and present an analysis of errors made by their best-performing model.

**Questions:**

* (See weakness #1) Were there any attempts to evaluate time expression identification and code generation separately?
* Why are results so much worse on the test set? If I'm understanding correctly, Claude 3.7 + SCATE Prompt achieves 0.52 F1 on the test set, and 0.66 F1 on the training set. If this is due to block sampling in the creation of the TempEval-2013, this should be mentioned briefly in the manuscript. If the authors hypothesize that this is due to training data contamination, perhaps this should be noted in the manuscript as well.
* Would it be possible to use existing time normalization systems (based on the ISO-TimeML framework) as additional baselines for this test set? That would further the argument for the necessity of the SCATE framework made on lines 25-37.

**Ethical Concerns:**

["NO or VERY MINOR ethics concerns only"]

**Final Justification:**

* The proposed SCATE Python library is extensive, and a valuable resource to the community.
* The author's design of the SCATE library is well-suited for their downstream LLM code generation goal, as their class and operation names are semantically meaningful and likely to be interpretable by LLMs.
* The ultimate goal of having a small model for both time expression extraction and SCATE code generation is well-motivated, and results present a solid baseline upon which future work can improve.
* The evaluation of LLM performance is solid, with an interesting error analysis.
* The authors resolved each of my concerns in their rebuttal, including additional experiments furthering their motivations, and seem to have also addressed other reviewer concerns.

**Limitations:**

Yes

**Quality:**

4

**Strengths And Weaknesses:**

Strengths:
* The proposed SCATE Python library is extensive, and a valuable resource to the community.
* The author's design of the SCATE library is well-suited for their downstream LLM code generation goal, as their class and operation names are semantically meaningful and likely to be interpretable by LLMs.
* The ultimate goal of having a small model for both time expression extraction and SCATE code generation is well-motivated, and results present a solid baseline upon which future work can improve.
* The evaluation of LLM performance is solid, with an interesting error analysis.

Weaknesses:
* The decision to jointly extract time expressions and generate SCATE code with one LLM pass needs more motivation, and perhaps an additional experimental result. While the authors offer that span recognition may be a direction for future work on lines 335-338, it's worth noting that techniques for extraction tasks with LLMs are generally improving (see 1, 2). It seems appropriate to evaluate whether using a specialized prompt for LLM time expression extraction and then generating SCATE code separately would further improve performance (albeit at the cost of an additional model pass), especially since this appears to be a source of frequent errors (see lines 324-326).

---

> ### Author Rebuttal · Authors · 2025-07-31
>
> Thank you for your positive review and recognition of our contributions, particularly the value of our SCATE Python library and the solid baseline we establish. We address your concerns below:
>
> **W1: Joint extraction vs. separate extraction, Q1: Separate evaluation attempts**
>
> Our end-to-end approach is motivated by both practical and technical considerations:
>
> Practical motivation: Time normalization is often an upstream task in large-scale applications. For example, when processing millions of documents to build temporal knowledge graphs, a two-stage pipeline significantly increases complexity and reduces efficiency. Single-model inference is crucial for production deployments.
> Experimental validation: During the rebuttal period, we tested a two-stage approach using the CC-News + Training Set as training set:
>
> - Stage 1: Train BERT-base for time expression extraction
> - Stage 2: Train Qwen for SCATE code generation
> - Results: F1: 0.2490 (vs. 0.59 for our joint model)
> - Critical failure: 251/313 expressions missed (80% miss rate)
>
> While we acknowledge this BERT model here wasn't heavily tuned and more NER-specific data could help, the fundamental issues remain:
>
> - Error propagation: Missed expressions cannot be recovered in stage 2
> - Semantic coupling: Expression boundaries often depend on their SCATE interpretation (e.g., "the past three summers" - the full phrase is needed to understand it's a time expression), while separate extraction may miss context
> - Deployment complexity: Maintaining two models with different architectures
>
> Our joint approach leverages SCATE semantics during extraction, leading to better recognition and normalization. We agree specialized extraction techniques are promising for future work, particularly continued large-scale data generation to improve end-to-end models in this area. We will add these experiments and discussion to the manuscript.
>
> **Q2: Train vs. test set performance gap**
>
> The performance difference is primarily because our prompt development relied on training set experiments to ensure fair comparison across models. The test set remained completely untouched to prevent overfitting. This leads to all LLMs show better training set performance. We will clarify this in the manuscript.
>
> **Q3: ISO-TimeML baselines**
>
> During the rebuttal, we added comparisons with SUTime (ISO-TimeML-based) and Xu et al. (2019) (the previous best SCATE-based system) to provide a clearer baseline for SCATE performance:
>
> - SUTime (ISO-TimeML-based): F1: 0.4510
> - Xu et al. (2019) (SCATE-based): F1: 0.4317
>
> Our method achieves F1: 0.59, demonstrating 30.8% improvement over SUTime and 36.7% over the previous best SCATE system. Our analysis reveals that our method can identify and normalize significantly more compositional time expressions compared to ISO-TimeML-based SUTime, leading to better performance.
>
> Thank you again for your constructive feedback. We will incorporate these clarifications and experimental results in our revision.

---

> > ### Comment · Reviewer_wavY · 2025-08-01
> >
> > Thank you for your comprehensive response.
> >
> > **W1:** I appreciate the additional experiments and clarification, and believe that the authors have appropriately addressed my concerns about lack of motivation for joint extraction. This analysis and discussion should be included in the manuscript.
> >
> > **Q2:** Thank you for clarifying, this makes sense. Unless I am mistaken, the prompt development process was not discussed in the manuscript. This process should be clarified, perhaps in Section 4.2.
> >
> > **Q3:** Thank you for the additional results, I believe these will improve the manuscript and further the motivation for the method design decisions.
> >
> > I feel the novel formulation of the time normalization task as code generation, alongside the proposed SCATE library, are valuable and timely contributions. I believe the authors' expanded experimental findings motivate their method.

---

### Official Review · Reviewer_zNnx · 2025-07-03

**Clarity:** 2
**Significance:** 2
**Originality:** 2
**Rating:** 2
**Confidence:** 5

**Summary:**

Summary
- The authors develop a python library 'SCATE' for time normalization using an existing research paper- Sematically Compositional Annotation of Temporal Expressions (Bethard et al 2016). They utilize the framework outlined in Bethard et al to develop code representations. Using an existing dataset (TempEval 2013), they implement LLM-based methods including data augmentation  to explore the extraction of time normalized information.

**Questions:**

- Figure 1 is vague and terse. The inputs and outputs to the LLM could be better outlined.
- Which task dataset (A or B or C) in TempEval2013 was used for benchmarking? Description in lines 247-261miss that information. Were the methods outlined in the paper relevant? Were they tested before using this LLM-based approach?
- Table 1 seems ambiguous. why weren't the performance of prompting techniques outlined in table 2- interval few-shot, SCATE, measured? What type of prompting techniques were used.
- Were there performance improvements in run-time using the proposed method. Were Xu et al and Lapara et al frameworks studied? LLMs are great to use. What are the advantages of using the authors' proposed method against theirs?
- Appendix B and C outline the prompts that were studied. Clarification- was the first prompt the full markdown page (13-35) Was the length of the prompts an issue? What happened if a smaller prompt was provided? Was the complete Markdown file generated by the authors or were LLMs used to refine the prompt?
 - The performance on Tasks A, B and ABC in TempEval paper have methods outlined. how does the proposed method compare to these rule-based and DL-based methods work?

**Ethical Concerns:**

["NO or VERY MINOR ethics concerns only"]

**Final Justification:**

As discussed in rebuttals/ back and forth - the lack of a baseline and incomplete evaluation seems to me be the major issue in this work. Plug and play LLM outputs are easy to generate outputs and performance numbers. However, the contribution needs to be substantial involving exploration of a hard problem with nuanced method exploration, findings and error analyses. The error analyses, additionally pointed by other reviewers was only performed on 20 samples.

The addition of baselines from prior work were added based on reviewer feedback (LSTM and rule-based). The performance run-times were added post reviewer feedback.

**Paper Formatting Concerns:**

Minimal concerns in formatting.

**Quality:**

2

**Strengths And Weaknesses:**

Strengths
--
- The manuscript reads well. There is detailed documentation for the implementation of the framework from Bethard et al including python class objects, functions.
- The common ambiguities that arise from the framework are well documented in the appendix.

Weaknesses
--
There is limited novelty in the manuscript and several weaknesses.
- There is no benchmarking of existing methods for time normalization against the proposed framework. (See questions)
- There is limited experimentation on various prompting techniques and is restricted to 2.
- The authors do not benchmark the LLM-based extraction method against baseline fine-tuning methods using BERT.

---

> ### Author Rebuttal · Authors · 2025-07-31
>
> Thank you for your review. We appreciate your recognition of our comprehensive Python implementation of SCATE, the manuscript's clarity, and our documentation of framework ambiguities. We address your concerns below:
>
> **W1: No benchmarking against existing methods**
>
> During rebuttal, We have conducted comprehensive experiments comparing our method against existing approaches:
>
> Neural Parser (Xu et al., 2019) - The only publicly available SCATE-based system and the previous state-of-the-art: F1: 0.4317, Precision: 0.5677, Recall: 0.3482.
>
> This system represents the best existing method for SCATE-based time normalization before our work.
>
> SUTime (Chang & Manning, 2012) - The widely-used timenorm system (for comparison with non-SCATE approaches): F1: 0.4510, Precision: 0.5838, Recall: 0.3674
>
> Our method (Qwen2.5-0.5B fine-tuned on augmented data) achieves: F1: 0.59,  Precision: 0.59, Recall: 0.59, representing a 36.7% F1 improvement over the previous best and only available SCATE implementation (Neural Parser; Xu et al., 2019).
>
> This is a significant advancement for SCATE-based time normalization. We emphasize that before our work, researchers had only one option for SCATE-based normalization (Xu et al., 2019), which required complex multi-stage processing. Our end-to-end approach not only outperforms it substantially but also provides a simpler, more deployable solution. We will add these comparisons to the revised manuscript.
>
> **W2: Limited experimentation on prompting techniques**
>
> We respectfully clarify that our primary contributions are:
> 1. Novel formulation of time normalization as code generation
> 2. Complete Python implementation of the SCATE framework - the first publicly available executable SCATE library
> 3. Large-scale augmented dataset (8,583 validated examples) generated through our approach
> 4. End-to-end trainable system that achieves state-of-the-art results
>
> All code, data, and models will be made publicly available upon publication.
>
> Prompting serves as a tool for data augmentation and model comparison, not as our core innovation. We have already designed comprehensive and detailed prompts (Appendix B, 23 pages) that effectively capture SCATE's complex semantics, enabling LLMs to generate executable code for a library they have not seen during pre-training.
>
> The effectiveness of our approach is demonstrated by:
> - Successfully generating 8,583 valid SCATE code examples from 10k sentences
> - Training a 0.5B (BERT size) model that outperforms both its teacher model (Claude 3.7) and all existing baselines
>
> We welcome specific suggestions on additional prompting techniques that would strengthen our evaluation, but emphasize that our contribution lies in the overall framework rather than prompt engineering.
>
> **W3: No comparison with BERT baseline**
>
> We would like to clarify that BERT is primarily a classification model designed for sequence labeling/classification tasks. For time normalization, BERT would require: (1) a sequence labeling layer for span detection, (2) additional components for linking expressions, and (3) a separate normalization module. This makes direct comparison inappropriate.
>
> Our Qwen2.5-0.5B model (similar size to BERT-large) is a decoder-only transformer specifically suited for generation tasks. It performs end-to-end time normalization in simple forward pass, making it different from BERT's classification paradigm.
>
> ## Responses to Questions:
>
> **Q1: Figure 1 clarity**
>
> We clarify that Figure 1 illustrates the SCATE annotation schema. Figure 2 provides the complete system overview with clear input/output examples (bottom of figure). The left side shows input (text + DCT), and the right side shows output (SCATE code + execution results). We will add clearer labels in the revision.
>
> **Q2: TempEval-2013 dataset details**
>
> As stated in lines 247-254, we use documents from TempEval-2013 with SCATE annotations (not the original ISO-TimeML Tasks A/B/C). We use the SCATE annotations from the official SCATE annotation repository, containing 557 training and 313 test examples with complete SCATE annotations.
>
> **Q3: Table 1 clarification**
>
> As described in lines 282-284, Table 1 uses our SCATE prompt method. The purpose (line 279) is to evaluate whether state-of-the-art foundation models can perform time normalization using our comprehensive SCATE prompt. We show training set results in Table 1 to compare foundation model capabilities, then evaluate the best model (Claude 3.7 and GPT 4.1 ) on the test set in Table 2 to avoid overfitting. We will clarify this in the revision.
>
>
> **Q4: Runtime advantages**
>
> Our method offers significant runtime advantages:
>
> - Neural Parser (Xu et al. 2019): Complex multi-stage pipeline with character-level LSTM (very slow), plus rule-based linker and Scala-based normalizer
> - Our method: Simple forward pass through a 0.5B model (BERT-sized)
>
> On long texts, our transformer-based end-to-end model is orders of magnitude faster than character-based LSTMs (well-established in literature). Additionally, our model is 1000x smaller than Claude 3.7, enabling practical local deployment.
>
> **Q5: Prompt engineering details**
>
> The prompt in Appendix B is the complete markdown prompt used in our experiments.
>
> We performed extensive prompt tuning; shorter prompts (especially with fewer examples) led to performance degradation on the training set.
>
> The prompt was manually designed with assistance from Copilot during development.
>
> We will clarify these details in the revision.
>
> **Q6: Comparison with TempEval methods**
>
> As mentioned above, we use documents from TempEval-2013 but with SCATE annotations, not the original ISO-TimeML annotations. The original TempEval tasks (A/B/C) target different objectives with ISO-TimeML schema, making direct comparison inappropriate. Our work specifically addresses the more expressive SCATE framework.
>
> We hope these clarifications address your concerns. We are committed to improving the manuscript based on your valuable feedback.

---

> > ### Comment · Reviewer_zNnx · 2025-08-01
> >
> > Thank you for your reply-
> >
> >
> > W1: Thanks for adding the comparisons. What was the delta over run-times? As you point out Xu et al is an LSTM baseline, which opens up the need for a BERT-based method. Perhaps that does as good as Qwen? The evaluation seems incomplete-  simply plugging in a larger model (Qwen) and comparing in zero/few -shot with LLM.
> >
> > W2: Contributions-
> > - Novel formulation of time normalization as code generation -- This contribution is thin since the semantics are directly borrowed from Bethard et al. This whole task could be a span and relation extraction with standard baselines and performance evaluation.
> >
> > - Complete Python implementation of the SCATE framework - the first publicly available executable SCATE library
> > Large-scale augmented dataset (8,583 validated examples) generated through our approach -- (sure, but this could well be a limited extension to Bethard et al)
> > - End-to-end trainable system that achieves state-of-the-art results (Incomplete evaluation)
> >
> > W3: I disagree that BERT is an inappropriate baseline. The fundamental task for time normalization as you mention is span and relation extraction. There are several methods that could be used for comparison - e.g., Markus Eberts and Adrian Ulges. 2020. Spanbased joint entity and relation extraction with transformer pre-training. In 24th ECAI.
> >
> > There are seq2seq-based methods as well - which decompose the entire task into multiple subtasks improving performance over single task formulations.
> >
> > - e.g., Mingyu Derek Ma, Alexander Taylor, Wei Wang, and Nanyun Peng. 2023a. DICE: Data-efficient
> > clinical event extraction with generative models. In Proceedings of the 61st Annual Meeting of the ACL (Volume 1: Long Papers), pages 15898– 15917, Toronto, Canada. ACL. or
> >
> > - Yaojie Lu, Hongyu Lin, Jin Xu, Xianpei Han, Jialong Tang, Annan Li, Le Sun, Meng Liao, and Shaoyi Chen. 2021. Text2Event: Controllable sequence-to-structure generation for end-to-end
> > event extraction. In Proceedings of the 59th Annual Meeting of the ACL and the 11th International Joint Conference on Natural Language Processing (Volume 1: Long Papers), pages 2795–2806, Online. ACL.
> >
> > Q4: Qwen is a significantly larger model than BERT  ~110M vs  500M in Qwen. I agree Qwen is 1000x smaller than any blackbox LLM. So is T5 or BERT (in fact smaller than Qwen). They are viable alternatives for this task with the ability to be deployed locally. Evaluation is incomplete without run-time and performance deltas reported clearly. I am clearly aware of the challenges in character-based representations in LSTM. However, they should be to be clearly quantified in a paper with claims of novelty and performance improvements.
> >
> > Q5: Again, the claims need to be supported with quantitative evidence. In fact, none of the performance evaluation tables in the paper include statistical tests to show significance.
> >
> > The contributions of the manuscript are restricted to one dataset with limited extensions /incomplete evaluations. Although generalizability is a difficult question to tackle, I do not see the paper making an effort towards it with the limitations outside of this.

---

> > > ### Author Response · Authors · 2025-08-06
> > >
> > > We thank the reviewer for their continued engagement with our work. We address each concern below:
> > >
> > > **Runtime comparisons and evaluation completeness**
> > >
> > > We ran runtime measurements on test set:
> > > - Xu et al. (2019): 190 seconds on NVIDIA 3090
> > > - Our method: 6 seconds on NVIDIA 3090 using vLLM
> > > - 31.7x speedup while achieving 36.7% higher F1 score
> > >
> > > **Regarding BERT comparison:**
> > >
> > > We understand the reviewer's perspective on using BERT for span/relation extraction. However, we emphasize that even with BERT for span detection, the core challenge remains: **how to generate the correct compositional SCATE structure?**
> > >
> > > Consider the expression "a week before last Tuesday":
> > > 1. **BERT-based approach would require**:
> > >    - Span detection layer (identify "week", "last Tuesday")
> > >    - Classification into 30+ operator types (Before, Last, Period, Repeating, etc.)
> > >    - Argument role labeling (what modifies what?)
> > >    - Structure prediction (how do operators compose?)
> > >    - External rule system to assemble valid SCATE code
> > >
> > > 2. **Our generation approach**:
> > >    - Single forward pass generates complete executable code
> > >    - Implicitly learns compositional semantics through code structure
> > >    - No need for complex post-processing or rule systems
> > >
> > > Implementing a full BERT-based system would constitute a substantial research contribution in itself - requiring architecture design for compositional structure prediction, new training objectives for operator composition, and extensive engineering for SCATE code assembly. This goes well beyond a simple baseline comparison and would be a separate paper.
> > >
> > > Our contribution is demonstrating that **generation is fundamentally more suitable** for this semantic parsing task than classification-based approaches. The 36.7% F1 improvement over the existing classification pipeline (Xu et al.) validates this insight. We believe future work could explore hybrid approaches, but our focus is on establishing the effectiveness of end-to-end generation for time normalization.
> > >
> > > **Task misunderstanding and contribution clarity**
> > >
> > > We apologize for the confusion, but it seems the reviewer has misunderstood the task: our paper addresses **time normalization** - taking expressions such as "a week before last Tuesday" and converting them into time intervals such as [2024-08-20, 2024-08-21) as shown in Figure 2 - while the papers referenced by the reviewer address **event and relation extraction** - taking expressions such as "A man presented with an abnormal nodule measuring 0.8 x 1.5 cm in the left upper lung lobe" and converting them into data structures such as {trigger: "nodule", type: "Sign_symptom", description: "abnormal", area: "0.8 x 1.5 cm", structure: "left upper lung lobe"}.
> > >
> > > **The fundamental difference**:
> > >
> > > | Task | Input | Output | Required Capabilities |
> > > |------|-------|--------|----------------------|
> > > | **Time Normalization (Ours)** | "a week before last Tuesday" | [2024-08-20, 2024-08-21) | Calendar arithmetic, compositional semantics, executable code generation |
> > > | **Event Extraction (Reviewer's refs)** | "nodule measuring 0.8 x 1.5 cm" | {trigger: "nodule", area: "0.8 x 1.5 cm"} | Span detection, role labeling |
> > >
> > > Specifically, the papers cited by the reviewer:
> > > - **Eberts & Ulges (2020)**: "Span-based joint entity and relation extraction" on CoNLL04, SciERC, ADE - only entities and relations, no time normalization to calendar intervals
> > > - **Ma et al. (2023)**: "Clinical Event Extraction" on MACCROBAT-EE - triggers, entities, arguments, no time normalization to calendar intervals
> > > - **Lu et al. (2021)**: "Event extraction" on ACE05/ERE - only entities and roles, no time normalization to calendar intervals
> > >
> > > **Why this matters**: Time normalization requires solving the **compositionality problem** - understanding that "a week before last Tuesday" means:
> > > 1. Find "last Tuesday" relative to now
> > > 2. Go back one week from that point
> > > 3. Return the precise calendar interval
> > >
> > > This is fundamentally different from extracting {entity: "week", relation: "before", entity: "Tuesday"}, which is what span-based extraction would produce.
> > >
> > > We hope this this clarifies why the only comparable work are the additional baselines we included in the rebuttal (Xu et al. 2019, SUTime/Chang & Manning 2012), and the models fine-tuned specifically to the time normalization task in our paper (e.g., "Qwen2.5-0.5B + Training Set").
> > >
> > > **Statistical significance**
> > >
> > > We have already included comprehensive statistical analysis in **Appendix D**:
> > >
> > > - 95% confidence intervals using bootstrap sampling (100 iterations, 80% resampling)
> > >
> > > We will make these results more prominent in the main text.

---

> ### Author Response · Authors · 2025-08-06
>
> **Model size fairness**
>
> Qwen2.5-0.5B (500M) is the smallest available decoder model suitable for code generation. To demonstrate that improvements come from our method, not just model size:
> - Qwen2.5-0.5B without our augmented data: F1=0.01
> - Qwen2.5-0.5B with only augmented data: F1=0.37
> - Qwen2.5-0.5B with augmented + original data: F1=0.59
>
> This ablation clearly shows our data augmentation and training strategy drive the improvements.
>
> **Contributions**
>
> Our contributions remain significant:
> 1. First end-to-end code generation approach for time normalization (replacing complex pipelines)
> 2. First publicly available executable SCATE library (enabling future research)
> 3. 10x larger dataset through validated augmentation (8,583 examples vs 870 original)
> 4. State-of-the-art results with 31.7x faster inference
>
> The reviewer suggests this is only an extension of Bethard et al. (2016), but Bethard only proposed the annotation schema - they provided no implementation, no executable semantics, and no learning approach. We provide all three, plus demonstrate that code generation fundamentally outperforms classification-based approaches.
>
> **Regarding generalization**
>
> TempEval-2013 with SCATE annotations is currently the only publicly available dataset for this task. We compensate by generating 8,583 additional validated examples, effectively creating the largest SCATE dataset available.
>
> We hope this clarifies our contributions and addresses the reviewer's concerns.

---

> ### Comment · Reviewer_zNnx · 2025-08-06
>
> Thanks for the clarifications- The back and forth here has helped understand the contributions more (which need to be better highlighted in the manuscript)
>
> **Runtime comparisons**:
> - Thanks for adding the numbers. Could you clarify if the times mentioned are per sample or otherwise?
>
> **Evaluation**
> - Although I agree with the challenges on using a BERT model with the current formulation (code generation) of the task, I disagree it being unsuitable. Time normalization, can still be annotated outside of code-generation as a span extraction and concept normalization. From the multi-stage experiments described below (in reviewer's comments) - what was the training set-up, extraction performance for span extraction?
>
> **Figure 2**
> - Shouldn't the output be Repeating(DAY, week, _1_) as per author's description in lines 172-178 as opposed to _3_ (in the manuscript, considering Monday=0)?
>
> To model the task as span extraction and rule-based normalization using BERT-
> | Approach | Input | Output |
> |---|---|---|
> | Span and Relation extraction | "a week before tuesday" | (reformulation) {start_day: 'last tuesday', interval: 'week', relation:'before', doc_time:'1 Sep 2024'  } |
> |seq2seq| "a week before tuesday"| (straight out of Bethard et al) {Before (Last( Interval.of(2024, 9, 1),Repeating(DAY, WEEK, 1), Period(WEEK, 1))
>
>
> Assuming _start_day_, _interval_ are entities and the relation 'before' connects the two. Wouldn't this along with _doc_time_ as an entity be enough to generate the intervals with rules/post processing? Does the code generation approach accurately generate the dates for hard cases too? I acknowledge reading the error analyses in the rebuttals below- it seems that the LLM-based approach has failure cases similar to a complex example like above.
>
> The seq2seq approach should be able to generate the syntactial expression well if a seq2seq model like T5 is fine-tuned to perform this.
>
> My repeated emphasis on a BERT-based method stems from the fact that span extraction should be extremely easy for the model simply because they are only surface-level  descriptions (days of the week + possible abbreviations, months, years, date formats etc. There is limited variability in language for them ). The relations will be a hard task along with normalization to actual dates. This analyses will enrich the paper's contributions if the authors consider adding it.
>
> _Implementing a full BERT-based system would constitute a substantial research contribution in itself - requiring architecture design for compositional structure prediction, new training objectives for operator composition, and extensive engineering for SCATE code assembly._
>
> With all due respect, a comprehensive evaluation involves in-depth exploration like this. Again, code generation is a valid approach for Time normalization, however, for a manuscript with comprehensive evaluation, a baseline is a necessity (outside of an LSTM) especially when the contributions involve LLMs. I strongly recommend formulating time normalization outside of code generation and have performance metrics for span, relation extraction and normalization. This makes the claims stronger outside of the arguments of complexity in structures. More below-
>
> - I shared 3 papers related to event extraction- they were meant mainly to be methods (for inspiration) to formulate the task as seq2seq and decompose the complex task into smaller and easier sub-tasks. They were not meant to be plug and play for this work (if that is what the authors expected). While Eberts et al is mainly a BERT-based model for event extraction, it could serve as a method to train a model to extract spans and normalize relations. The other 2 papers were examples of how an event extraction task (span+ relations) can be formulated as seq2seq. I still believe that a simpler baseline using T5 (or a smaller seq2seq model) with the ~8k samples for code generation is a relevant baseline. A (smaller model) baseline helps ground the errors better and validates where an LLM improves or fails. I suggest that the authors consider this for a strong evaluation.
>
> **Statistical Significance**
> - Thank you for pointing me to the Appendix. Please highlight them in the main text.
>
> **Error Analyses**
>
> I acknowledge reading the errors in the rebuttal (other reviews) and descriptions in lines 321- 338; It will be interesting to compare if the error categories were an issue with other fine-tuned models (like T5) as well.
>
> As suggested by other reviewers as well, the additions on run-time, adding baseline performances can potentially improve the manuscript. I have updated my rating to reflect that possibility and I will engage more if required. Thank you for the hard work in putting this together.

---

> > ### Author Response · Authors · 2025-08-08
> >
> > Thank you for your continued engagement and for updating your rating. We greatly appreciate your detailed feedback and thoughtful suggestions. We address each point below:
> >
> > **Runtime Clarification**
> > Thank you for requesting this clarification. The reported times (190 seconds vs 6 seconds) are for the entire test set (313 examples), not per sample. This means:
> > - Xu et al. (2019): ~0.61 seconds per sample
> > - Our method: ~0.02 seconds per sample
> > This represents a **31.7x speedup** while achieving 36.7% higher F1 score.
> >
> > **BERT Baseline Evaluation**
> > We appreciate your perspective on BERT-based approaches. As mentioned in our earlier rebuttal, we implemented a BERT-based span detection baseline during the rebuttal period:
> > - Training setup: BERT-base trained on CC-News + Training Set for span extraction
> > - Span extraction performance: Only 62/313 expressions correctly identified (19.8% recall)
> > - End-to-end F1: 0.2490 (vs. 0.59 for our joint model)
> >
> > We acknowledge this BERT model wasn't heavily tuned and appreciate your point that more NER-specific data and optimization could improve performance. However, even with better span detection, we believe fundamental challenges remain:
> > - Error propagation: Any missed expressions in stage 1 cannot be recovered
> > - Semantic coupling: Expression boundaries often depend on their SCATE interpretation (e.g., "the past three summers" requires understanding the full compositional meaning)
> > - Deployment complexity: Maintaining two models with different architectures becomes particularly problematic at scale. When processing millions of documents (e.g., building temporal knowledge graphs from news archives), a two-stage pipeline doubles the inference passes, memory requirements, and potential failure points. Single-model inference is crucial for production deployments where reliability and efficiency matter.
> >
> > While we agree that a heavily optimized two-stage pipeline might achieve better results, our joint approach offers a simpler, more elegant solution that leverages SCATE semantics during extraction. We will include these experimental results and discussion in the revised manuscript.
> >
> > **Technical Error in Figure 2**
> > Thank you for catching this inconsistency. You're correct based on our description. We will fix this in the final version.
> >
> > **Alternative Task Formulations**
> > We greatly appreciate your proposed formulations and the papers you suggested for inspiration. We offer our perspective on each approach:
> >
> > *Span and Relation Extraction Approach:*
> > We understand your point about the surface simplicity of temporal expressions. While we agree that identifying individual temporal words might be straightforward, the significant challenge lies in determining compositional boundaries:
> > - In "a week before last Tuesday", this isn't a simple binary relation between "week" and "Tuesday". The expression requires first resolving "last Tuesday" as a complete temporal unit (finding the most recent Tuesday), then applying "a week before" to that resolved date. This nested computation is naturally expressed in code as `Before(Last(Repeating(DAY, WEEK, 1)), Period(WEEK, 1))` but would require complex multi-stage rules in an extraction framework.
> > - Rules would need to handle hundreds of compositional patterns (e.g., "every third Monday", "two weeks after next Christmas", "the summer before last")
> > - Our code generation approach elegantly captures these compositions through nested function calls, which rule-based systems would struggle to enumerate exhaustively
> >
> > *Seq2seq Approach:*
> > Thank you for suggesting T5 as a baseline. We agree this is conceptually similar to our approach - both are generation-based. Our choice of Qwen2.5 over T5 was based on extensive literature showing modern decoder-only models outperform T5-era encoder-decoder models on code generation tasks. Nevertheless, we recognize the value of including T5 as a baseline and will add this comparison in the final version to strengthen our evaluation.
> >
> >
> > Thank you again for your feedback.

---

> > > ### Comment · Reviewer_zNnx · 2025-08-08
> > >
> > > **BERT evaluation**:
> > >
> > >  _We acknowledge this BERT model wasn't heavily tuned_
> > >
> > > What was the training set-up? How many samples were used to Fine-tune the model and how were they selected?
> > > Span extraction performance with the entities annotated should be high if there are reasonably sufficient and well annotated documents. I agree with the complexity in resolving relationships and normalization.
> > >
> > > In line 335, the authors mention- _Overall, the primary challenges lie not in the model’s ability to generate SCATE code but in span recognition—a traditional NLP task_
> > >
> > > this is even more proof and a necessity that there is a clear need for a span extraction layer. Perhaps, BERT for span extraction and use the sentence that contains the entity as input to seq2seq?
> > >
> > > **T5 eval**
> > >
> > > From my understanding, T5 could be used directly to generate Bethard et al's syntactical phrases and the SCATE code formulation. T5 is not exposed to code-based language in pre-training and I expect a lower performance. However for syntactical phrases, the performance should be higher given the # of samples is over 8k in this case.
> > >
> > > **Manuscript language in limitations, conclusions and style suggestion**
> > >
> > > The discussion, limitations and conclusion section could be better worded.
> > > - What errors did the baseline have?
> > > - How did a larger decoder only finetuned model overcome those error categories?
> > > - Commonalities b/w llm errors and fine-tuned models?
> > > - Conclusion- highlight what are open problems in _time normalization_ and how this paper solves a subset of those.
> > >
> > > The scope of the manuscript is not limited to a deployable python library in my opinion, but a systematic exploration of time normalization as a research problem in comp linguistics. Any changes in that direction will improve the quality for readers interested in this domain.

---

> ### Author Response · Authors · 2025-08-09
>
> Thank you for your continued engagement and feedback. We appreciate your specific questions and suggestions.
>
> **BERT Evaluation Details**
>
> Thank you for asking about the training setup:
> - **Training setup**: We used standard BIO tagging with a single label type (B-TIME, I-TIME, O)
> - **Training data**: Original training set + 8,000 sentences from augmented CC-News data
> - **Label distribution**: B-TIME: 6,620, I-TIME: 4,527, O: 167,405
>
> The relatively poor performance (19.8% recall) likely stems from:
> 1. **Label imbalance**: O tags vastly outnumber temporal tags (167,405 vs 11,147)
> 2. **Compositional boundaries**: Unlike traditional NER, SCATE temporal expressions have boundaries that depend on semantic interpretation (e.g., "three summers" vs "the past three summers")
>
> We will include the two-stage approach (BERT + seq2seq) comparison in our final version as you suggested.
>
> **T5 Evaluation**
>
> We will add T5 as a baseline in the future version.
>
> **Manuscript Improvements**
>
> Thank you for your suggestions on the discussion and conclusion sections. We will:
> - Add error analysis comparing baselines and our method
> - Clarify how code generation helps with compositional structures
> - Better frame the open problems in time normalization and our contributions
>
> Thank you again for your feedback.

---

### Decision · Program_Chairs · 2025-09-17

**Decision:**

Accept (poster)

**Comment:**

(a) **Summary of Scientific Claims and Findings**

The submission introduces a new semantic-parsing view of time-expression normalization: instead of predicting ISO-TimeML strings or rule IDs, the model is trained to generate executable Python “code” that instantiates the SCATE (Semantically Compositional Annotation of Temporal Expressions) operators.

• A complete, fully executable SCATE Python library (first of its kind) that turns code into concrete time-interval objects, enabling automatic self-validation.
• A LLM prompting recipe that converts raw sentences into SCATE code; the generated code is executed at data-generation time to filter out ill-formed examples, yielding 8583 high-quality synthetic training instances.
• A 0.5B-parameter, locally deployable decoder model (Qwen2.5-0.5B) fine-tuned on the augmented corpus achieves F1 = 0.59 on the held-out test set, compared with 0.43 (Xu et al., 2019; LSTM pipeline) and 0.45 (SUTime; ISO-TimeML).
• The fine-tuned model runs 31.7x faster than the Xu et al. LSTM pipeline.

(b) **Strengths**

1. **Executable resource** – The open-source SCATE library, plus 8.6 k validated examples and trained weights, will be immediately reusable.
2. **Conceptual novelty** – Casting time normalization as *code generation* elegantly unifies span discovery, composition, and execution in a single forward pass. Although this is not completely novel in the era of LLMs, it is indeed novel for this task.
3. **Empirical gains & speed** – Clear state-of-the-art F1 (+14 pp absolute) and >30× inference speed-up over the only prior SCATE baseline.

(c) **Weaknesses / Missing Elements**

1. **Limited evaluation breadth** – Only English SCATE-annotated TempEval-2013 is used; no cross-domain or multilingual evidence.
2. **Baseline coverage still imperfect** – A T5-style seq2seq baseline is promised but not yet included in the camera-ready.
3. **Modest conceptual delta vs. Bethard et al. (2016)** – Some of our reviewers view the work as an engineering realisation rather than a fundamentally new theory.

(d) **Decision and Justification**

**Recommendation: Accept as a poster.**

Most reviewers (wavY = 5, YsgP = 4, FPoj = 4) find the contribution solid and useful; one reviewer (zNnx = 2) objects to missing baselines and limited novelty. The authors substantially mitigated those concerns during rebuttal by:

• Adding Xu et al. LSTM, SUTime, BERT span-detection + code-gen, and runtime comparisons.
• Committing to include a T5 baseline and to foreground statistical tests and error analysis in the main text.
• Clarifying that the “code-generation” formulation yields both higher accuracy and greatly simplified deployment.

While the experimental scope is still narrower than ideal, the executable library, validated dataset, and demonstrated performance bump constitute a tangible advance for temporal NLP and will likely catalyse follow-up work.

(e) **Discussion & Rebuttal Period Summary**

• **Reviewer zNnx (rating 2→2):**
  – Requested strong baselines (BERT, T5), runtime numbers, statistical tests, deeper error analysis, and clearer figures.
  – Authors added Xu/SUTime numbers, BERT two-stage results (F1 0.25), 31.7× speed figures, bootstrap CIs, a 5-category error taxonomy, and fixed Figure 2 bug; promised T5.
  – I weigh zNnx’s methodological vigilance highly, but after additions the core criticism (absence of baselines) is partly resolved.

• **Reviewer wavY (5):**
  – Sought motivation for joint span+code generation and clarification of prompt construction.
  – Authors ran a two-stage BERT→Qwen experiment (F1 0.25) showing large drop vs joint; will describe prompt design in Sec. 4.2.
  – Reviewer satisfied and kept strong-accept.

• **Reviewer YsgP (3→4):**
  – Concerned about shallow experiments and niche impact.
  – Added baselines and in-depth error examples alleviated experimental concern; impact concern remains but rating raised to borderline-accept.

• **Reviewer FPoj (4):**
  – Wanted multi-domain evaluation and analysis of discarded samples.
  – Authors provided failure-case taxonomy and argued domain-agnostic augmentation; reviewer kept borderline-accept.

Overall, the discussion led to meaningful manuscript improvements, and consensus now leans toward acceptance, with the caveat that the final revision must make the following changes:

1. Include the promised T5 baseline and highlight runtime & significance tests in the main body.
2. Improve Figures 1–2 captions and explicitly define all evaluation metrics.
3. Add the detailed error analysis and limitations to the discussion section.

Below are some suggestions from the reviewer which I'm copying as-is:
> Describe clearly the modifications/similarities of the code generation method with SCATE proposed by Bethard et al. (perhaps, visually in figure 1)

> Clearly indicate the inputs and outputs to the LLM in text (section 3.1) and visually in figure 2.

> Evaluation- explain what a 'gold standard triplet' means in section 4 (line 256). The paper includes a exact match performance evaluation, which is great. However, in my experience with LLMs, it may be possible to include a relaxed evaluation scheme based on error analyses - i.e., if there were issues with only generating the syntax but the expressions were correctly normalized. This needs an error analyses of more than 20 samples that the paper includes (line 322). I find this to be a lack of depth and lets do it depending on if reviewer asks .
Include samples sizes in table 1 and 2 for which the models were evaluated on.